# People Make Better Edits: Measuring the Efficacy of LLM-Generated Counterfactually Augmented Data for Harmful Language Detection

**Indira Sen[1,2], Dennis Assenmacher[3], Mattia Samory[4],**
**Isabelle Augenstein[5], Wil van der Aalst[1],** and **Claudia Wagner[1,3]**
[1]RWTH Aachen University [2]University of Konstanz
[3]GESIS - Leibniz Institute for Social Sciences [4]Sapienza University of Rome
[5]University of Copenhagen
`indira.sen@rwth-aachen.de,mattia.samory@uniroma1.it`
`augenstein@di.ku.dk,wvdaalst@pads.rwth-aachen.de`
`{dennis.assenmacher,claudia.wagner}@gesis.org`

## Abstract

NLP models are used in a variety of critical social computing tasks, such as detecting sexist, racist, or otherwise hateful content. Therefore, it is imperative that these models are robust to spurious features. Past work has attempted to tackle such spurious features using training data augmentation, including Counterfactually Augmented Data (CADs). CADs introduce minimal changes to existing training data points and flip their labels; training on them may reduce model dependency on spurious features. However, manually generating CADs can be time-consuming and expensive. Hence in this work, we assess if this task can be automated using generative NLP models. We automatically generate CADs using Polyjuice, Chat-GPT, and Flan-T5, and evaluate their usefulness in improving model robustness compared to manually-generated CADs. By testing both model performance on multiple out-of-domain test sets and individual data point efficacy, our results show that while manual CADs are still the most effective, CADs generated by Chat-GPT come a close second. One key reason for the lower performance of automated methods is that the changes they introduce are often insufficient to flip the original label.[1]

**Warning:** This paper has instances of hateful and sexist language to serve as examples.

## 1 Introduction

For a given text with an associated label, a counterfactual example is obtained by *making minimal changes to the text in order to flip its label*, i.e., converting a hateful tweet into a non-hateful tweet. Table 1 shows an original tweet and its counterfactual pairs from different generation mechanisms.

| Data type | Example |
|---|---|
| original | I do not like female engineering teachers |
| manual CAD | I do not like ~~female~~ engineering teachers |
| polyjuice CAD | I do not like female managers teachers |
| ChatGPT CAD | I do not have a preference for female or male engineering teachers. |
| Flan-T5 CAD | I ~~do not~~ like female engineering teachers |

Table 1: Different types of counterfactuals generated from a sexist original instance. The highlighted part indicates what has been changed from the original.

Counterfactual examples have the interesting property that, since they were generated with minimal changes, they differ from the original instance in one aspect only—typically only the NLP task we want to model (also called the "construct"), is different. We can train ML models on the pair of data points consisting of the original and counterfactual examples to make the models focus on the NLP task rather than spurious training artifacts.

Previous work has shown that integrating counterfactually augmented data (CADs) into the training data of NLP models can improve their performance on Out-of-Domain (OOD) datasets for the same construct (Kaushik et al., 2020; Samory et al., 2021; Sen et al., 2021). However, previous work focused on manually generated CADs, which can be expensive and time-consuming to create. Automating the CAD generation process can reduce this manual labor while allowing OOD generalizability.

In this work, we compare manual CADs to automated CADs generated from Polyjuice (Wu et al., 2021), ChatGPT,[2] and Flan-T5 (Chung et al., 2022) as training data for harmful language detection models, specifically sexism and hate speech detection. With recent advances in generative NLP approaches, specifically Large Language Models (LLMs), their dominance in many NLP bench-

---

[1]The dataset of automatically generated CADs and their properties, our code, as well as our trained or finetuned models are available here: `https://github.com/Indiiigo/automatedCAD`

[2]https://openai.com/blog/chatgpt

marks (Chung et al., 2022) and their use in social computing generative tasks (Josifoski et al., 2023; Veselovsky et al., 2023), we expect these generation techniques to produce high-quality CADs, especially recent variants of Large Language models which can be accessed via prompt-based interfaces like ChatGPT and Flan-T5.

This work aims to shed light on (i) **RQ1: the extent to which different automated CAD generation methods are capable of generating efficient CADs that boost model performance**. We then use the complementary perspective of information-theoretic formulations of dataset difficulty (Ethayarajh et al., 2022) to obtain the 'difficulty' of manual and automated CADs, i.e., how hard it is for a model family to learn them or how much useful information they entail for learnability; We then explore (ii) **RQ2: the properties of CADs making them effective as training data** - i.e., those that help improve model performance; Using instance-level difficulty scores, we study links between various properties of CADs (such as the semantic and lexical distance from the original instance or their label) to surface the properties of effective CADs.

We find that models trained on a combination of original and manual CADs outperform models trained on just originals on several OOD datasets for both sexism and hate speech detection. While ChatGPT CADs are better training data than other automated CADs, they still fare worse than manual CADs. In terms of CAD properties, the generation mechanism plays a strong role in the efficacy of CADs — Flan-T5 and Polyjuice CADs have high difficulty, indicating too little usable information is available from them for a model to learn. By studying various properties of CADs, we discover that this is likely due to the fact that they make insufficient changes to flip the label and end up creating mislabeled CADs. Our results show that while mixing manual and automated CADs can improve OOD generalizability, we need manual vetting to ascertain the automated CADs' labels instead of using a fully automated pipeline.

## 2 Data and Methods

### 2.1 Training and Test Datasets

Following past work (Kaushik et al., 2020), we differentiate in-domain (ID) from out-of-domain (OOD) data. Training and test datasets are summarized in Table 3 and 4, respectively. Unlike previous work that was limited to single OOD datasets,

| Model Type | Model Name | Trained On |
|---|---|---|
| non -counterfactual | OG | 100% original data (no CADs) |
| counterfactual | mCAD | original data, manual CADs |
| | aCAD$_{PJ}$ | original data, automated Polyjuice CADs |
| | aCAD$_{GPT}$ | original data, automated ChatGPT CADs |
| | aCAD$_{FT}$ | original data, automated Flan-T5 CADs |
| | amCAD | original data, manual and automated CADs |

Table 2: Different types of models trained for each model architecture.

we evaluate our models on a wider range of OOD datasets. All our datasets are in English language.

### 2.2 CAD Generation Methods

**Manual CAD generation.** We re-use CADs manually generated in past research (Samory et al., 2021; Vidgen et al., 2021) to compare against automated CADs. All manual CADs are obtained by asking annotators to make 'minimal changes to flip the label' for the corresponding construct, either generated by trained crowdworkers (Samory et al., 2021) or expert annotators (Vidgen et al., 2021). Samory et al. (2021) generate several CADs per original, but to be consistent across both constructs, we randomly sample one CAD-original pair for sexism.

**Automated CAD generation.** We generate automated CADs using the following methods:

**Polyjuice** (Wu et al., 2021) produces general-purpose CAD, which may or may not flip the label w.r.t. a certain construct. It utilizes a fine-tuned GPT-2 (Radford et al., 2019) model to create counterfactually augmented data from input sentences by using eight different control codes (e.g., negation, shuffling, lexical) that are provided as a condition to the model to control the output generation. For each original instance, we generate five counterfactuals with all codes. However, not all codes provided a response, potentially because it could not be applied to create a coherent counterfactual from the original instance. Indeed, one of the control codes, 'shuffle,' returned null output for all original instances. The other seven had varying coverage, 'lexical' being the most frequent.

**ChatGPT** is described as a 'sibling' model to InstructGPT (Ouyang et al., 2022), designed to respond to prompts, by training a GPT-3.5 model through Reinforcement Learning through Human Feedback (Christiano et al., 2017). It has been used

| construct | Original | | Manual CAD | | Polyjuice CAD | | ChatGPT CAD | | Flan-T5 CAD | |
|---|---|---|---|---|---|---|---|---|---|---|
| sexism | S | nS | S | nS | S | nS | S | nS | S | nS |
| Samory et al. | 1244 | 1610 | - | 648 | - | 691 | - | 1242 | - | 1141 |
| hate speech | H | nH | H | nH | H | nH | H | nH | H | nH |
| Vidgen et al. | 6503 | 5759 | 5759 | 6503 | 3381 | 3277 | 5679 | 6409 | 5130 | 6065 |

Table 3: **In-domain datasets and Counterfactually Augmented Data (CADs) used for training models, and the distribution of instances across classes [S = sexist, nS = non-sexist, H = hate, nH = not hate].** For sexism, Samory et al. asked crowdworkers to generate non-sexist counterfactuals from sexist original instances but not the other way around. Therefore, for fair comparisons, we also only incorporate non-sexist automated CADs for sexism detection for all other counterfactual models.

| construct | ID | | OOD1 | | OOD2 | | OOD3 | | OOD4 | | HC | |
|---|---|---|---|---|---|---|---|---|---|---|---|---|
| sexism | Samory et al. | | Rodriguez-Sanchez et al. | | Guest et al. | | Kirk et al. | | | | Röttger et al. | |
| | S | nS | S | nS | S | nS | S | nS | S | nS | S | nS |
| | 534 | 690 | 1636 | 1800 | 699 | 5856 | 4854 | 15146 | | | 373 | 136 |
| hate speech | Vidgen et al. | | Qian et al. (R) | | Basile et al. | | Qian et al. (G) | | Mandl et al. | | Röttger et al. | |
| | H | nH | H | nH | H | nH | H | nH | H | nH | H | nH |
| | 471 | 464 | 14614 | 19162 | 1260 | 1740 | 5256 | 17067 | 1267 | 5738 | 2563 | 1165 |

Table 4: **Test sets for both constructs [S = sexist, nS = non-sexist, H = hate, nH = not hate] .** Qian et al. contribute two hate speech datasets, one sourced from Reddit and the other from Gab. We count these as separate datasets indicated by (R) and (G), respectively. For sexism, we only have three OOD datasets and HC, hence OOD4 is blank.

in several NLP classification and generation tasks with strong results (Ziems et al., 2023; Qin et al., 2023). We accessed ChatGPT via the OpenAI API.

**Flan-T5** (Chung et al., 2022) is a prompt-based LLM similar to ChatGPT, but the underlying model is openly available. Flan-T5 was created by instruction finetuning several model families, including T5 (Raffel et al., 2020). Due to computational constraints, we use the medium-sized model in the Flan-T5 family (Flan-T5 large).

The CADs generated via these methods for sexism and hate speech are summarised in Table 3. While we generate multiple CADs per original for the automated methods, we randomly pick one for each method to pair with the original. We first note uneven numbers of CADs for different CAD generation methods, e.g., 691 Polyjuice CADs non-sexist CADs vs. 1242 ChatGPT non-sexist CADs. This is because not all CAD generation techniques have the same coverage — ideally, for 1244 sexist original instances, we would have 1244 non-sexist CADs from all techniques, including manual generation. However, for 553 sexist instances, Polyjuice returned null values; similarly, ChatGPT returned default responses, while Flan-T5 returned the exact same text as the original for some original instances. Therefore, we only keep those non-null outputs that are at least a single character different from the original instance and not a default re-

sponse from ChatGPT.[3] The LLM hyperparameters and prompts and the steps taken to detect Guardrail LLM responses are in the Appendix (Section A and H). *We assume all CADs have the opposite label of their original counterparts.* We believe this to be reasonable since the prompts given to ChatGPT and Flan-T5 explicitly instructed this (as it did to people generating manual CADs), and one of the benefits of automation is reducing manual labor (including having to ascertain CADs' labels).[4]

### 2.3 Model Architectures

We finetune RoBERTa (Liu et al., 2019) models, Flan-T5 large (treated as sequence-to-sequence problem), and SVM models with Fasttext embeddings (Joulin et al., 2016) on the data sets described in Table 3.

**Baselines.** We use few-shot labels from ChatGPT ($FS_{GPT}$) and Flan-T5 large ($FS_{FT}$), and the Perspective API's (Lees et al., 2022) toxicity endpoint ($P_{Tox}$) as baselines that we have not trained.[5]

---

[3] Samory et al. (2021) and Vidgen et al. (2021) also manually vet the manual CADs for sexism and hate speech, respectively and remove invalid CADs, explaining why the manual approach also does not have 100% coverage.

[4] Wu et al. (2021) state that Polyjuice does not flip the label between 40-63% of the time, depending on the task, and manual vetting is needed to determine the labels. However, simulating a regime without manual vetting of the automated CADs gives us a conservative estimate of automated CAD.

[5] The full prompts used to generate few-shot labels and the preprocessing done on the outputs of all three baselines to obtain binary sexist/hateful vs. non-sexist/non-hateful labels

## 3 Experimental Setup

### 3.1 Harmful Language Detection Models.

For each model architecture (e.g., RoBERTa), we have various types of models based on whether they have been trained on CADs or not, as well as which type of CAD. Table 2 summarizes the six model types, while Table 3 summarizes the data used for training the different models. In addition to the four types of models trained on different CADs, one on the manual CADs and three on the different types of automated CADs, respectively, we include an additional model trained on a mixture of manual and automated CAD. We do this to simulate a 'wisdom-of-(automated)-crowds' scenario, where we also train on CADs selected randomly (called *amCAD*) from the total corpus of CADs generated by different methods, stratified by their availability, e.g., Polyjuice CADs are fewer compared to the others because they could not be generated for all original instances. All model parameters and hyperparameters are included in the Appendix (Section A).

**Metrics.** We use *macro F1* to assess overall performance.

### 3.2 Sampling the Training Data

For hate speech, all counterfactual models are trained on 50% original data and 50% CADs from the respective sources, stratified by label. For sexism detection, there are only non-sexist manual CADs (obtained by changing the sexist CADs), and therefore the counterfactual models only include 25% CADs. For a fair comparison, we also inject only the non-sexist automated CADs into the sexism models trained on automated CADs.

Since some CAD generation methods do not provide CADs for all instances (cf. Section 2.2), to make our results consistent and rule out confounds due to differential training data sizes, we **ensure that all models are trained on equal amounts of randomly sampled (balanced) data.** Therefore, for the training datasets of the various CAD models, we sample original and CAD pairs from all available CADs for that method. To state an example, if ChatGPT cannot augment instance $i$, but Flan-T5 can then $i$ is included in the training set of aCAD$_{FT}$ but not in the training set of aCAD$_{GPT}$. While the training sets are then different across augmentation strategies, this is done intentionally — this is closer to real-world settings where we would

use high coverage augmentation strategies which are by extension more diverse.

### 3.3 Dataset Difficulty Analysis.

For RQ2, we need instance-level scores of individual CADs to investigate which of their characteristics make them effective training examples. We use *V-Information* (Ethayarajh et al., 2022) and its instance-level counterpart *Pointwise V-Information (PVI)* to score each dataset instance w.r.t. the training set distribution, including CADs. V-information denotes the ease with which a model family can predict the label of inputs and can be measured using the predictive V-Information framework (Xu et al., 2020), a generalization of Shannon information accounting for computational constraints. PVI extends V-information as Pointwise Mutual Information extends Shannon Information, indicating the learnable information from an instance w.r.t. a given distribution. In summary, PVI scores indicate the amount of information that the input (in this case, a text) provides to the harmful language detection model about the label (sexism or hate speech), compared to the absence of input. The higher the PVI score of an instance, the easier it is to learn, i.e., the more useable information it has for a model.[6] Low PVI scores, especially negative ones, entail difficult cases with little useable information, often indicating mislabeled instances. Thus, PVI scores can help us gauge the properties of CADs with more usable information. Therefore, we reuse the *amCAD* counterfactual RoBERTa model (the one containing original data and randomly selected manual and automated CADs) and estimate the PVI scores of the training data. We then use the CADs' PVI scores as the dependent variable in an ordinary least squares (OLS) regression model with the CAD properties (described in Section 4.2) as independent variables. Interpreting the beta coefficients of these variables allows us to estimate the association between these properties and the difficulty of CADs.

## 4 Results

### 4.1 RQ1: How does model performance change when trained on automated CAD?

For each type of model described in Table 2, we train models over five runs and test them on the test

---

are included in the Appendix (Section H and G).

[6]It is unclear if easy-to-learn instances are useful for out-of-domain generalizability. However, they improve model convergence and contain more information than instances with negative PVI scores, making them more effective.

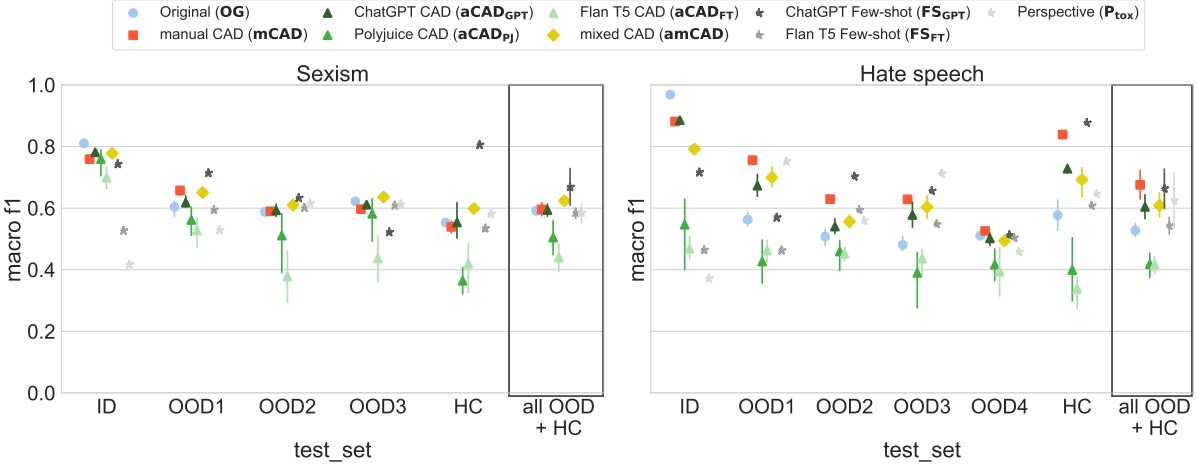

Figure 1: **The performance of different types of RoBERTa models for detecting sexism and hate speech on different types of test sets, including the macro average on all out-of-domain test sets, called "all OOD+HC" [RQ1].** Note that $aCAD_{GPT}$ is a finetuned RoBERTa model on the CADs generated by ChatGPT, while $FS_{GPT}$ denotes the few-shot classification labels from ChatGPT. Models trained on manual CADs ($mCAD$) have low in-domain performance compared to non-counterfactual models but higher performance out-of-domain for virtually all OOD datasets. The manual CADs are better than the automated ones, but CADs from ChatGPT come close to matching their performance. Manual CADs are most effective for hate speech, while for the sexism models, a mix of manual and different types of automated CADs yields the best OOD performance among the finetuned models. (Statistically significant using McNemar's test, see Appendix D) Few-shot labels from ChatGPT ($FS_{GPT}$) perform well, especially for sexism and for the hate check datasets, however, with high variance across different datasets.

sets described in Section 2.1, summarized in Figure 1. For both constructs, the RoBERTa models outperformed the other models — substantially for the SVM models, and to a minor extent for the finetuned Flan-T5 models. Therefore, we only include the results of the RoBERTa models in the main paper. The results of the finetuned Flan-T5 and SVM models, which showed similar trends but lower overall scores, are in the Appendix (Section C).

*For both sexism and hate speech detection RoBERTa models we see that models trained on manual CADs (*mCAD*), ChatGPT CADs (*aCAD_{GPT}*), or a mixture of manual and automated CADs (*amCAD*) outperform all other finetuned models on the OOD datasets, indicating better generalization capabilities.* Models trained on automated CADs from Polyjuice and Flan-T5 (*aCAD_{PJ}* and *aCAD_{FT}*, respectively) have poor OOD performance, while models trained on ChatGPT CADs (*aCAD_{GPT}*) were better in OOD datasets but still not as good as models trained on manual CADs, especially for hate speech detection, indicating that manual CADs outperform automated CADs as training data in aiding OOD generalizability.

Among the baselines, the best-performing finetuned model is typically better than the Perspective API and Flan-T5 few-shot labels (the only exception being OOD3 for hate speech, where Perspective is better). However, few-shot labels from Chat-GPT are quite competitive and often outperform the finetuned RoBERTa models, especially for hat-echek — however, this high performance could be due to data contamination (Chang et al., 2023; Aiyappa et al., 2023; Augenstein et al., 2023). *Indeed, for sexism, ChatGPT performs poorly compared to the finetuned models on OOD3 (Kirk et al., 2023), a recent dataset that followed the launch of ChatGPT, providing more indication that the older datasets may have been a part of ChatGPT's training data.* Hence, we suggest caution in comparing ChatGPT's results against the finetuned models. We also note that the best performing finetuned models — *amCAD* for sexism and *mCAD* for hate speech outperform the Toxicity API, a model widely used for measuring toxicity when training a custom model is not possible (Ribeiro et al., 2021; Papasavva et al., 2020; Rajadesingan et al., 2020). Our results indicate that models trained on CADs are suitable alternatives to black-box options such as the Toxicity API. We therefore facilitate access to these models via the Huggingface Hub.[7]

To summarize overall OOD performance, we compute the average macro F1 of all models, av-

---

[7]https://huggingface.co/AutoCAD

eraged across all OOD datasets in Figure 1. *For hate speech, training on manual CADs improves OOD performance the most, while for sexism, models trained on a mixture of manual and automated CADs lead to the highest results.* These patterns are also statistically significant, as ascertained by the McNemar's test (see Appendix D for detailed results).

## 4.2 RQ2: What are the properties of individual effective CADs?

While the previous research question investigated the efficacy of CADs based on performance in OOD test sets, we now use a set of properties discussed in past literature and assess how these properties are linked with CADs' learnability.

**CAD Properties.** Based on past literature (Kaushik et al., 2020; Vidgen et al., 2021), we look into three properties of CADs — 1) **minimality** operationalized as the Levenshtein edit distance from the original instance on a token level, 2) **semantic similarity**, operationalized by SBERT (Reimers and Gurevych, 2019) cosine similarity score with the original, and 3) **edit types** based on the type of edits made; addition or deletion of negation, gender/identity word, and affect or emotion words (Sen et al., 2021). For sexism, we only include gender word edits, and for hate speech, we only include identity word edits to match the topical focus of each construct. We use various lexica to assess whether a CAD has a certain type of edit. To adjudicate if a gender and identity word was edited, we use a list of gender words,[8] and identity terms, respectively.[9] For negation, we reuse the list compiled by Ribeiro et al. (2020) for making NLP test suites, and finally for affect words, we use a lexicon of affect words (Hu and Liu, 2004).

Additionally, we also consider the 4) **source of CADs** (the generation method) and the 5) **CAD labels**, but only for hate speech, as we only have one-sided CADs for sexism.

**Manual vs. Automated CADs.** Our results for RQ1, inline with past work, show that training on manual CADs can promote OOD generalizability, while Sen et al. (2021) demonstrate that certain types of manual CADs are better based on edit type. In this work, we study a wider range of properties for both manual and automated CADs, and find

that there is some variability within the automated CADs — in terms of edit distance, Flan-T5 and Polyjuice have shorter edits (average token-level edit distance of 2.6/2.6 and 1.71/1.67, respectively for sexism/hate speech), while ChatGPT has much more (11.64/14.5). Manual CADs sit somewhere in between (2.42/6.68). We see a similar pattern for semantic similarity — Flan-T5 and Polyjuice have higher semantic similarity to the original (0.91/0.86 and 0.9/0.89 respectively, for sexism/hate speech), while ChatGPT CADs are more semantically distant (0.67 for both sexism and hate speech). Manual CADs again sit in between (0.81/0.73), *indicating that automated CADs make either too many or too few changes compared to manual CADs.* Regarding types of edits, we do notice similarities between manual and ChatGPT CADs, especially for sexism, where both favor the deletion of gender words in making sexist instances non-sexist. These properties are described in the Appendix (Section E).

**Information-Theoretic Data Valuation.** As described in 3.3, we reuse the *amCAD* model and obtain PVI score for its training data containing both original data and CADs. However, to assess if PVI scores reflect CADs' abilities to improve OOD performance, we first compare the mean PVI scores of the OOD test sets for models trained on original data vs. models trained on CADs and original data (i.e., the *amCAD* model).[10] We find that models trained on a combination of original data and CADs have a higher average PVI score for the OOD datasets — models trained on original data have an OOD mean PVI score of -0.05 (sexism) and -0.19 (hate speech). In contrast, models trained on a mixture of original data and counterfactuals have a score of 0.1 (sexism) and 0.17 (hate speech). *The increase in average PVI scores indicates that incorporating CADs in our training data reduces the dataset difficulty of OOD data and makes it easier to learn compared to a setting where only original data points are used, further corroborating the efficacy of CADs in boosting out-of-domain performance.* The average PVI scores on the OOD test sets are in the Appendix (Section F, Figure 4).

Having established that training on CADs im-

---

[8]https://github.com/uclanlp/gn_glove/tree/master/wordlist

[9]obtained by combining the keywords used in (Khani and Liang, 2021) and hatebase (https://hatebase.org/)

[10]Ethayarajh et al. (2022) estimate PVI scores by training a model on a subset of the data sampled from the same distribution. In this light, estimating PVI scores for OOD data departs from Ethayarajh et al. (2022)'s assumption. We argue that such scores, even if not directly interpretable as PVI, still convey the informativeness of an example with respect to the model used to estimate PVI and in relation to its training data's distribution.

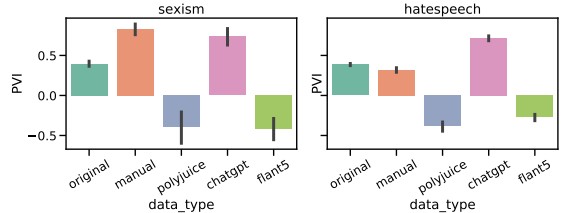

Figure 2: **Distribution of PVI scores for the training set containing original data and CADs.** For both tasks, CADs from Polyjuice and Flan-T5 have the lowest PVI scores indicating they are the hardest-to-learn.

proves generalizability by making OOD datasets easier to learn, we try to assess which types of CADs are most difficult to learn for these models. To that end, we obtain PVI scores of the training set containing original data and CADs from a model trained on a combination of original data and CADs (the *amCAD* model).[11] Figure 2 shows the distribution of PVI scores by type of data, including the type of CAD. Ethayarajh et al. (2022) find the lowest PVI scores to be often indicative of mislabeled instances. Indeed, for manual CADs, the labels were vetted, so they are less likely to be mislabeled, while the same cannot be concluded for the automated CADs. We assess the most difficult CADs (those with the lowest PVI scores), such as the Flan-T5 CAD "Men are *less* willing to take risks than women" (emphasis on the changed token) obtained from the original instance "Men are *more* willing to take risks than women" with a PVI score of -1.91. From this example, it is clear that the change is trivial, involving the exchange of comparative words; a change that does not flip the sexist label at all. It is possible that many of these low PVI-scored instances do not change the original labels and are therefore mislabeled instances. *This also indicates that automated CADs require a manual check to establish their correct label for them to be used as training data.* The top five most difficult (low PVI-scored) CADs and easiest (high PVI-scored) CADs for both constructs can be found in the Appendix ( Tables 10 and 11 respectively.)

**Regression Analysis.** To better understand which properties of CADs impact their learnability, we train ordinary least squares (OLS) regression models to predict PVI scores of CADs (the dependent variable), one for sexism and one for hate speech, with the following independent variables —

|  | Dependent variable: | |
| --- | --- | --- |
|  | $PVI_{sexism}$ | $PVI_{hate\ speech}$ |
| edit_distance | -0.051* | 0.012** |
| cosine_similarity | -0.074*** | -0.008* |
| type_.chatgpt | 0.018 | 0.060*** |
| type_flant5 | -0.287*** | -0.102*** |
| type_pj | -0.295*** | -0.127*** |
| gender_add | -0.248*** | |
| gender_del | 0.139*** | |
| label:identity_add | | 0.099*** |
| label:identity_del | | -0.018 |
| Observations | 498 | 4,606 |
| $R^2$ | 0.484 | 0.288 |
| Adjusted $R^2$ | **0.473** | **0.286** |
| *Note:* | *p<0.1; **p<0.05; ***p<0.01 | |

Table 5: **OLS models predicting the PVI scores of CADs for sexism and hate speech detection with the beta coefficient for the significant dependent variables.** The full regression table with all variables and other statistics is included in the Appendix (Table 12)

edit distance, cosine similarity, CAD source, edit type, and the label feature for hate speech, i.e., whether the CAD is hateful or not.[12] For hate speech where we have both hateful and non-hateful CADs, we also include interactions between the CAD label and the identity word addition and deletion edit type; we do so since the CAD label could be strongly connected to the presence of identity words — hateful CADs would be more likely to have identity terms. The increased association of identity terms and the hateful class label could drive unintended false positive bias (Sen et al., 2022; Dixon et al., 2018; Nozza et al., 2019). Since the CAD label and CAD source are categorical features, we one-hot encode them and use the positive class (hateful instances) and manual CADs as reference variables. Edit types are modeled as binary (1 indicates the presence of the corresponding edit, irrespective of how many such edits were done for an instance). Table 5 summarizes the two OLS model, one for sexism and one for hate speech.

We first note that the adjusted R² values for the sexism and hate speech models explain 47.3% and 28.6% of the variability, respectively. The moderate fit indicates that, especially for hate speech, while additional factors might help explain the efficacy of the CADs further, these scores can still provide informative insights about the role of different CAD characteristics, i.e., the dependent variables.

**Interpreting the Regression Results: Sexism.** First, the edit type also plays a role in the diffi-

---

[11]as described by Ethayarajh et al. (2022), we only train the model for two epochs and use a dev set (splitting the training data 80-20) to avoid overfitting when obtaining the PVI scores.

[12]For sexism, no label feature is needed since the CADs in sexism are all non-sexist

culty of CADs for sexism detection; gender word deletions are significantly associated with easier-to-learn CAD — possibly, since removing gender terms makes instances unrelated to sexism, making them more trivial to classify. In contrast, gender word additions are associated with harder-to-learn CADs since adding them likely leads to mislabeled instances where CADs still remain sexist. We see such instances in Flan-T5 and Polyjuice where one gender word is substituted for another without meaningfully removing the sexism (examples in the Appendix, Table 10). Indeed, being Flan-T5 or Polyjuice CAD is associated with lower PVI scores (-0.28, p < 0.01 and -0.29, p < 0.01, respectively). Cosine similarity is negatively correlated with ease (-0.074, p < 0.01) — *CADs are harder if they are semantically closer to the original instances.*

**Interpreting the Regression Results: Hate speech.** For hate speech, we see similar trends as sexism for semantically similar, Flan-T5, and Polyjuice CAD, i.e., that they are harder to learn. Notably, ChatGPT CADs are easier to learn than manual CADs (0.058, p < 0.01), *suggesting that manual CADs occupy the position Ethayarajh et al. (2022) described as 'ambiguous' instances with middling PVI scores that improve out-of-domain generalizability.* This is backed up by Figure 2 as well as the best-performing hate speech detection models for the OOD datasets being the models trained on manual CADs (RQ1, Figure 1). For the identity word edits, we see that adding identity terms is mediated by the CAD label — adding an identity word to make a hateful CAD is associated with higher PVI scores (0.1, p < 0.01), probably since adding an identity term is one of the common tactics to create hateful CADs — further reinforcing the association between hate speech and identity terms and exacerbating unintended false positive bias. Unlike sexism, edit distance is positively correlated with easiness (0.012, p < 0.05) for hate speech CADs.

**Human Validation of Automated CAD Labels.** To further validate our findings on the (lack of) label-flipping of automated CADs, we conducted a manual assessment of 100 automated CADs for each CAD generation technique for each construct, i.e., 300 CADs for sexism and 300 for hate speech. Two of the paper's authors manually and independently labeled the CADs without looking at the original instance. We calculated the initial agreement, measured using cohen's $\kappa$, which came out

to be 0.67 for sexism and 0.76 for hate speech, indicating substantial agreement (McHugh, 2012). Disagreements were then resolved based on discussions. We find that 57% of automated CADs for sexism and 70% for hate speech are mislabeled, i.e., they do not flip the original instances' label. Most mislabeled instances are from Flan-T5 and Polyjuice, but even ChatGPT does not flip the label 14% of the time for sexism, and 42% for hate speech.

To summarize, while training on CADs, in general, makes it easier to learn OOD datasets, there is some variance in the learnability of CADs based on their generation mechanism, edit type, and the validity of their labels. *We find that certain types of automated CADs are harder to learn due to their generation technique (Flan-T5 and Polyjuice), insufficient changes made to flip the label, and semantic closeness to the original instance.*

## 5 Related Work

Our work sits at the intersection of automated data augmentation and understanding the quality of NLP training data.

**Data augmentation** has received increased interest in the NLP community, specifically since the advent of LLMs (Feng et al., 2021). In this work, we focus on a specific type of augmented data: Counterfactually Augmented Data (CADs, also called Contrast sets by Gardner et al. (2020)). While some recent studies have used ChatGPT for data augmentation (Yoo et al., 2021; Dai et al., 2023; Møller et al., 2023), to the best of our knowledge, there is no prior work on using ChatGPT (or other prompt-based LLMs) to create label-flipping CADs to be used as training data.

**Counterfactuals in NLP.** Originally, CADs were created manually (Kaushik et al., 2020; Gardner et al., 2020; Samory et al., 2021). Recent work has looked into automated or semi-automated generation mechanisms (Ross et al., 2022; Wu et al., 2021; Atanasova et al., 2022, 2023). CAD generation techniques have been used to create challenging test sets (Madaan et al., 2021; Li et al., 2020; Ross et al., 2021; Robeer et al., 2021; Atanasova et al., 2022), for augmenting training data (Anuchitanukul et al., 2022; Howard et al., 2022), and for testing the faithfulness of explanations atanasova-etal-2023-faithfulness. While several researchers note the efficacy of CADs in boosting generalizability, other research has also shown that training on them can instead lead to more (Joshi and He,

2022) or others types of spurious correlations (Sen et al., 2022).

**Assessing Data Quality in NLP.** Recent approaches have looked beyond overall model performance to gain insights into training data efficacy. Swayamdipta et al. (2020); Bras et al. (2020) both introduce methods for observing the training dynamics of individual instances. Influence functions (Koh and Liang, 2017) observe local changes while Pezeshkpour et al. (2022) identify artifacts by measuring feature importance and instance attribution. Yet other work has extended Shannon information and pointwise mutual information for gauging dataset difficulty or the learnable information in datasets (Ethayarajh et al., 2022; Xu et al., 2020), especially interpretable instance-level difficulty scores, which we use in this work to find the properties of effective CADs.

## 6 Discussion and Conclusion

Counterfactually Augmented Data, inspired by the philosophy of causality (Kaushik et al., 2020), offers an elegant approach to improving te robustness and generalizability of automated classifier. However, they are expensive to generate manually and generating them automatically has many benefits. While past work has used LLMs like GPT-2 for generating CADs (Wu et al., 2021), to the best of our knowledge, there is still a lack of research on the direct comparison of manually generated CADs and automated CADs, especially those generated using instruction-based LLMs like Flan-T5 or GPT-3. The specific shortcomings of automated CADs are also understudies, especially for the use cases of harmful language detection. In this work, we take the first steps to show what the potentials and pitfalls of automated CADs are, and to surface which of their weaknesses future work needs to pay attention to.

Using two complementary perspectives — 1) overall performance on multiple out-of-domain datasets and 2) dataset difficulty metrics — we find that models trained on some types of Counterfactually Augmented Data (CAD) improve out-of-domain generalizability of both sexism and hate speech detection models. While automated CADs from ChatGPT boost model performance out-of-domain, they are typically not as effective as manual CADs (RQ1). We also try several baselines, including few-shot prompting of ChatGPT to label the OOD test sets. While ChatGPT's few-shot labeling performance often surpasses the models

trained on CADs, we cannot rule out data contamination; ChatGPT few-shot labels have high performance on all datasets predating 2022 but fall drastically for a recent sexism dataset from Kirk et al. (2023)

Next, using information-theoretic measures of dataset difficulty, we further unpack issues with automated CADs — Flan-T5 and Polyjuice CADs are generated without enough change to flip the label, while for hate speech ChatGPT CADs are too easy-to-learn compared to manual CADs, i.e., not ambiguous enough to fuel out-of-domain generalizability (RQ2). Our results indicate that CAD generation cannot be fully automated with current LLMs; they require manual checking. Furthermore, we find that CADs with identity words are associated more strongly with the hateful class, which could worsen unintended false positive bias by misclassifying non-harmful content. Nevertheless, we see some promise in combining manual and automated CADs indicated by the high performance of the *amCAD* models, which opens up intriguing possibilities for human-AI collaboration in counterfactual data augmentation; especially with prompt-based interfaces where automated text generation can be controlled to tailor CADs by maximizing the desirable properties we found in this work.

**Future work.** Building on the findings of our work, we can use human-in-the-loop setups combined with controlled text generation to design more precise automated counterfactuals that closely mimic the properties of manual CADs. Ideally, we can find paradigms where LLMs facilitate training data generation with human supervision.

Tangentially, counterfactual augmentation can not only be used to generate training data for training or conventional fine tuning of models — they can also be leveraged in instruction tuning, i.e., finetuning with explicit instructions about a particular task as well as labeled examples (Ouyang et al., 2022; Wei et al., 2022; Wang et al., 2023; Mishra et al., 2022). Indeed, by counterfactually and minimally perturbing instructions, and systematically analyzing the output, we can design ideal instruction tuning datasets, building on frameworks like 'Self-Instruct' (Wang et al., 2022).

## 7 Limitations

Our work is not free of limitations, including:

1. We focus on three approaches for generating CADs. We also tried other recent LLMs for

this purpose, however, their quality was either not as good (BLOOM with fewer parameters), or the models were too slow (Alpaca, GPT4All) or required more computational and hardware resources (LLaMa, bigger BLOOM and Flan-T5 models).

2. We also did not explore any prompt optimization techniques (Pryzant et al., 2023) for the LLM-based CAD generation methods. However, these techniques often rely on ground truth data and can be more effectively used for few-shot labeling, where gold-standard labels are available for some of the data being labeled. The same is not true for CAD generation, where there is no well-established label for quality. Indeed, to the best of our knowledge, no prompt optimization approaches are available for counterfactual generation. In this work, we show how dataset difficulty metrics can be used to find mislabeled CADs, and in the future, we could use these scores as an optimization objective for prompt improvement.

3. We assume that the out-of-domain datasets used in this work were independent of the in-domain training data and not partially or fully contained in the training data of our finetuned models (RoBERTa) and the models we used for few-shot labeling (ChatGPT and Flan-T5), however, this is difficult to guarantee especially for closed-source models like ChatGPT.

# 8 Broader Impact Statement

Our work contributes to assessing the validity of automated measurement of hate speech and sexism. Hate speech and sexism detection models are of paramount importance to both the NLP and broader (computational) social science research community since they are used for automated or semi-automated content moderation, as well as for measuring sexism and hate speech on social media platforms for informing research into vulnerable populations and policy-making. As we cannot create specific training sets for the different contexts where we want to measure sexism and hate speech, we need robust NLP models that can generalize. Our work shows how synthetic data augmentation, using counterfactually augmented data (CADs) can be used to improve this type of generalizability, directly contributing to making models more robust. We also show that despite recent advances in generative NLP, specifically LLMs, these techniques are still not at par with humans when it comes to creating CADs. However, by surfacing issues with current automated CADs, several future directions open up in using prompt engineering and Human-AI interfaces to improve the quality of CADs and NLP training data.

In general, we also find that effective CADs also tend to emphasize the association between identity terms and the hate/sexist label, which can fuel unintended false positive bias towards marginalized communities. To that end, we need further audits of these models to unearth the extent of these associations and their consequences. We also caution against fully automated deployment of these models in content moderation settings.

To generate synthetic NLP training data, we used LLMs to generate hateful content and found that while models like ChatGPT often gave guardrail responses and did not provide hateful instances for some original instances, in many cases they *did* for the exact same prompt. This type of stochastic LLM behavior highlights one of the risks of LLMs — they can and do generate hateful content. However, we only use such content to improve model robustness by training models on such data and explicitly labeling them as hateful. We do not endorse or encourage the use of LLMs to generate hateful content for any other purpose.

Finally, our sexism detection task models gender on a binary that does not account for sexism towards non-binary people or trans people. Unfortunately, almost all existing NLP datasets and models for sexism detection use a binary conceptualization of gender, which is exclusionary towards nonbinary and trans people. Our sexism detection models do, to some extent, model nuanced forms of sexism that plague gender minorities, including the essentialization of gender and gendered stereotypes. In the future, we hope to include conceptualizations of sexism that go beyond the gender binary.

## Acknowledgements

We thank the anonymous reviewers for their constructive and valuable feedback. We are also grateful to the members of the GESIS CSS department and CopeNLU for their helpful comments on a draft of this paper. Isabelle Augenstein's research was co-funded by the European Union (ERC, ExplainYourself, 101077481), as well as by the Pioneer Centre for AI, DNRF grant number P1. Views and opinions expressed are however those of the author(s) only and do not necessarily reflect

those of the European Union or the European Research Council. Neither the European Union nor the granting authority can be held responsible for them.

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

## Supplementary Materials

## A  Reproducibility

To ensure the reproducibility of our experiments, we are committed to sharing all of our training scripts and CAD datasets with the reviewers during the evaluation process and making them publicly available on GitHub upon publication. However, while we share all data with the reviewers, for the public repository, we cannot share the original data from past work (Samory et al., 2021; Vidgen et al., 2021) directly (referred to as OG) within our repository due to licensing restrictions. Instead, we will provide appropriate references to the OG data, allowing interested parties to access it through the designated sources. The same is applicable to some of the out-of-domain test sets in Table 4.

### A.1  Compute Infrastructure

All models were trained or finetuned on a machine with an AMD EPYC 7543 CPU and 20GB NVIDIA A100 MiG partition for the finetuning experiments and Flan-T5 CAD generation, and 40 GB for Flan-T5 few-shot labeling.

### A.2  Model Parameters

The number of parameters of all models used in this work is summarized in Table 6. The models include the sexism and hate speech models (SVM and RoBERTa), the CAD generation methods (Polyjuice, ChatGPT, Flan-T5), and the OLS regression models for predicting the PVI scores (Section 4.2).

| Model architecture | Number of parameters |
| --- | --- |
| Linear SVM | 300 (Fasttext dimensions) |
| RoBERTa (base) | 12-layer, 768-hidden, 12-heads, 125 Million |
| Polyjuice | 1.5 Billion (i.e., the same parameters as the GPT-2 model) |
| Flan-T5 | 0.8 Billion |
| ChatGPT (chatgpt 3.5 turbo) | 175 Billion |
| sexism OLS | 11 |
| hate speech OLS | 14 |

Table 6: **The number of parameters for different models used in this work.**

### A.3  Hyperparameter Optimization and Training Time

**Model Training.** We use the huggingface Transformers[13] and simpletransformers[14] libraries for model training. Note that we train or finetune 6 different models (Table 2) over 5 runs, each for different model architectures (RoBERTa and Linear SVM) for the two constructs (sexism and hate speech). Taken together, we have 120 models, which are then tested on different test sets — 31,724 test instances for sexism and 70,767 instances for hate speech. We then report the mean macro F1 on them with the standard deviations across runs in Figures 1 and 5. Given the many models we train and computational constraints, we could not do a full hyperparameter search for the RoBERTa models. Instead, we use early stopping to optimize the number of epochs using a dev set (by splitting the training data 80-20). Specifically, we start with 15 epochs with an early stopping patience of 5 and an early stopping delta of 0.01.

For the SVM models, which are computationally less intensive, we used five-fold cross-validation and grid-search for hyperparameter tuning. We executed the grid search with a range of (0.01, 100) with increments of 10, corresponding to 10 searches and with accuracy as the optimization metric. The hyperparameters we used for the models are as follows:

1. RoBERTa: learning rate (1e-6), batch size (32), maximum sequence length (512)

2. Flan-T5: learning rate (5e-4), batch size (64), dropout (0.05), steps (64k)

3. Linear SVM: C (0.01)

**CAD Generation and Few-shot Labeling.** We use the default hyperparameters from ChatGPT, i.e., do not specify any temperature or beam-search. For the other CAD generation and Few-shot labeling approaches, the hyperparameters are included in Table 7.

### A.4  Metrics

The evaluation metrics used in this paper are macro average F1 and False Positive Rate for RQ1. We

---

[13]https://huggingface.co/docs/transformers/index
[14]https://github.com/ThilinaRajapakse/simpletransformers

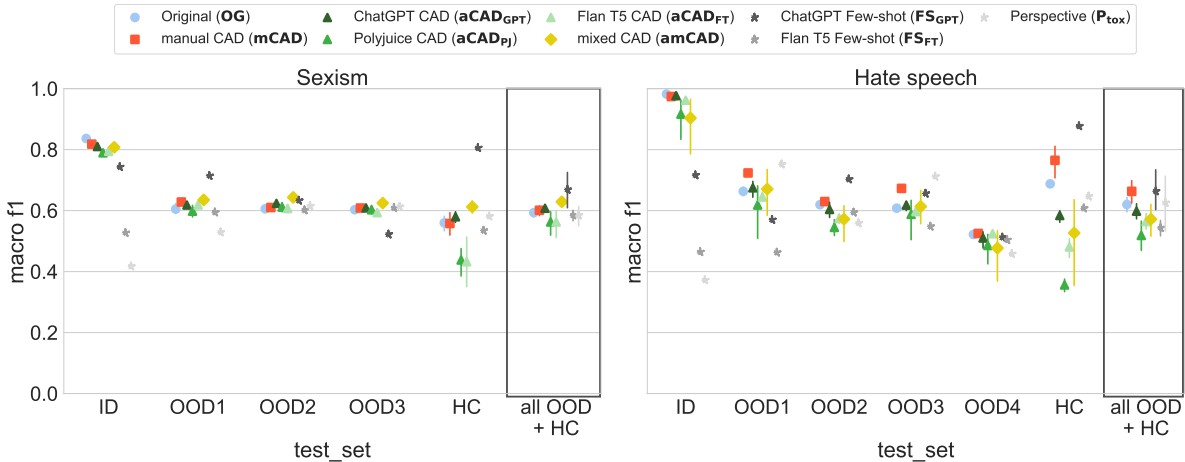

Figure 3: **The performance of different types of Flan-T5 models for detecting sexism and hate speech measured using macro F1.** For both sexism and hate speech, models trained on CADs outperform models trained on just original data. For sexism, manual, chatGPT, and a mixture of CADs perform best, while manual CADs are the best for hate speech.

| | **Hyperparameters** |
|---|---|
| **Polyjuice** | Perplexity: 10 |
| **ChatGPT** | default hyperparameters of chatgpt 3.5 turbo |
| **Flan-T5** | temperature: 1.5, num_beams: 10, max_length (only set for CAD generation and not few-shot labeling) 500 |

Table 7: **The hyperparameters for CAD generation (all three) and few-shot labeling (only applicable to ChatGPT and Flan-T5).**

used the sklearn implementation of these metrics.[15] For RQ2, we use V-Information.[16]

| construct | test_set | chatgpt | flant5 |
|---|---|---|---|
| **hatespeech** | HC | 0.058 | 0.104 |
| | ID | 0.13 | 0.041 |
| | OOD1 | 0.518 | 0.199 |
| | OOD2 | 0.073 | 0.154 |
| | OOD3 | 0.342 | 0.092 |
| | OOD4 | 0.067 | 0.047 |
| **sexism** | HC | 0.005 | 0.227 |
| | ID | 0.028 | 0.025 |
| | OOD1 | 0.032 | 0.038 |
| | OOD2 | 0.112 | 0.038 |
| | OOD3 | 0.053 | 0.073 |

Table 8: The proportion of LLM output that didn't conform to our requirements for the few-shot labeling.

[15]https://scikit-learn.org/stable/modules/generated/sklearn.metrics.precision_recall_fscore_support.html
[16]https://github.com/kawine/dataset_difficulty

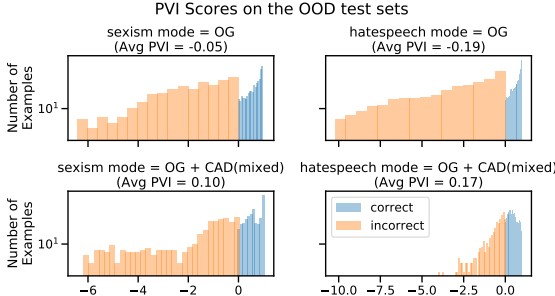

Figure 4: **Distribution of OOD test sets' PVI scores for models trained on datasets with and without CADs.** For both sexism and hate speech, models trained on original data and CADs have higher average PVI scores (V-information) on OOD test sets compared to models trained on just original data, implying that training on CADs makes the OOD dataset easier to learn.

## B Datasets

**In-Domain Datasets.** For the in-domain datasets Vidgen et al. (2021), we reuse the train-test splits as specified, except that the CADs are not a part of any test sets. For sexism, since no explicit train-test split is specified, we opt for a 70-30 train-test split.

**Out-of-Domain Datasets.** For the out-of-domain datasets and hate check, after downloading the data from the given source (included in the zipped data folder), we use the full data for testing. Hatecheck (Röttger et al., 2021) is a test suite of challenging instances for hate speech and abusive language detection. For hate speech, we use all instances, while for sexism we only include the instances targeting women (there are no instances

that target other men or nonbinary people).

## C Flan-T5 and SVM Results

We train counterfactual and non-counterfactual Flan-T5 and SVM models, in addition to the RoBERTa ones mentioned in the main paper. For the SVM models, the instances are encoded using Fasttext embedding, specifically the 300 dimensional word vectors, with character n-grams of length 5, a window of size 5 and 10 negatives, trained using CBOW.[17] We train these models using gridsearch and 5-fold cross-validation over five runs, with search space $C \in (0.01, 100)$, with increments of 10, and test them on the test sets reported in Table 4. We measure macro F1 (Figure 3 and Figure 5).

Compared to the RoBERTa models in 1, the fine tuned Flan-T5 models demonstrate better performance of CAD models trained on Flan T5 and Polyjuice CADs. However, the main tendencies seen in RoBERTa still hold, i.e, manual CADs are the best for hate speech, while a mix of manual and automated CADs is the best for sexism. Therefore, the results in our paper are not attributed to a particular model architecture. The unique ability of Flan-T5 in exploiting Polyjuice and Flan-T5 CADs, compared to RoBERTa, points to potential advantages in Flan-T5's model architecture which can be studied further in future work.

## D Significance Tests for RQ1

Given that we compare so many models, we did not report significance tests for all comparisons, however, for the main takeaways (1. mCAD is better than OG for both sexism and hate speech, 2. mCAD is better than aCAD$_{GPT}$ for hate speech, 3. amCAD is better than aCAD$_{GPT}$ for sexism) using the McNemar test (Dietterich, 1998) our findings are significant, except in one single run for sexism. We report these results in Table 9. Due to space constraints, we only report the results on the combined OOD datasets, but the results were also significant for the individual OOD datasets.

## E Manual vs. Automated CADs: Descriptive Summary

**Minimality.** Past instructions for generating CADs have emphasized minimality, and for a good reason — counterfactuals in the tradition of causal inference also seek "small" changes where only

| Construct | Run | McNemar | P value | Model 1 | Model 2 |
|---|---|---|---|---|---|
| sexism | 0 | 55.19 | 1.10E-13 | OG | mCAD |
| | 0 | 359.57 | 3.49E-80 | mCAD | aCAD$_{GPT}$ |
| | 0 | 126.16 | 2.84E-29 | amCAD | aCAD$_{GPT}$ |
| | 1 | 953.82 | 1.96E-209 | OG | mCAD |
| | 1 | 122.25 | 2.03E-28 | mCAD | aCAD$_{GPT}$ |
| | 1 | 99.04 | 2.47E-23 | amCAD | aCAD$_{GPT}$ |
| | 2 | 1376.57 | 2.59E-301 | OG | mCAD |
| | 2 | 842.23 | 3.56E-185 | mCAD | aCAD$_{GPT}$ |
| | 2 | 1.69 | 0.19 | amCAD | aCAD$_{GPT}$ |
| | 3 | 4279.53 | 0 | OG | mCAD |
| | 3 | 1311.52 | 3.54E-287 | mCAD | aCAD$_{GPT}$ |
| | 3 | 116.28 | 4.12E-27 | amCAD | aCAD$_{GPT}$ |
| | 4 | 1592.80 | 0 | OG | mCAD |
| | 4 | 3454.45 | 0 | mCAD | aCAD$_{GPT}$ |
| | 4 | 171.62 | 3.28E-39 | amCAD | aCAD$_{GPT}$ |
| hate speech | 0 | 954.18 | 1.64E-209 | OG | mCAD |
| | 0 | 11317.38 | 0 | mCAD | aCAD$_{GPT}$ |
| | 0 | 305.69 | 1.90E-68 | amCAD | aCAD$_{GPT}$ |
| | 1 | 2791.83 | 0 | OG | mCAD |
| | 1 | 8512.32 | 0 | mCAD | aCAD$_{GPT}$ |
| | 1 | 2207.61 | 0 | amCAD | aCAD$_{GPT}$ |
| | 2 | 345.41 | 4.22E-77 | OG | mCAD |
| | 2 | 13411.89 | 0 | mCAD | aCAD$_{GPT}$ |
| | 2 | 712.45 | 5.87E-157 | amCAD | aCAD$_{GPT}$ |
| | 3 | 4623.38 | 0 | OG | mCAD |
| | 3 | 3278.36 | 0 | mCAD | aCAD$_{GPT}$ |
| | 3 | 1573.69 | 0 | amCAD | aCAD$_{GPT}$ |
| | 4 | 559.08 | 1.33E-123 | OG | mCAD |
| | 4 | 17744.84 | 0 | mCAD | aCAD$_{GPT}$ |
| | 4 | 1944.13 | 0 | amCAD | aCAD$_{GPT}$ |

Table 9: Significance tests between 1) OG and mCAD models, 2) mCAD and aCAD$_{GPT}$, and 3) amCAD and aCAD$_{GPT}$ models using McNemar's significance test (RoBERTa models). All comparisons are statistically significant except for one run of sexism, which is colored red.

the causal variable of interest is changed (Pearl, 2014). We assess lexical minimality in this case, i.e., the number of tokens and characters changed in generating a CAD from its original counterpart.

Edit distance is one proxy for minimality, and we compute the word-level edit distance between individual original instances and their CADs from different sources (manual, polyjuice, chatgpt, and Flan-T5) using Levenshtein distanfce from the string2string library. [18]

We plot the distribution of the edit distances in Figure 7 and find that ChatGPT CADs tend to make the most changes while Polyjuice makes the least. Flan-T5 exhibits similar patterns as Polyjuice, while manual CADs are not as minimal as them, but more so than ChatGPT CADs.

**Semantic similarity.** Instructions for generating not only encourage minimality but also suggest adhering to the same topic and content of the original message to the extent possible. While such a property is somewhat difficult to quantify, we use semantic textual similarity computed with sentence embedding distances as a proxy for topical and con-

---

[17]https://fasttext.cc/docs/en/crawl-vectors.html

[18]https://github.com/stanfordnlp/string2string

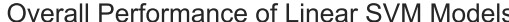

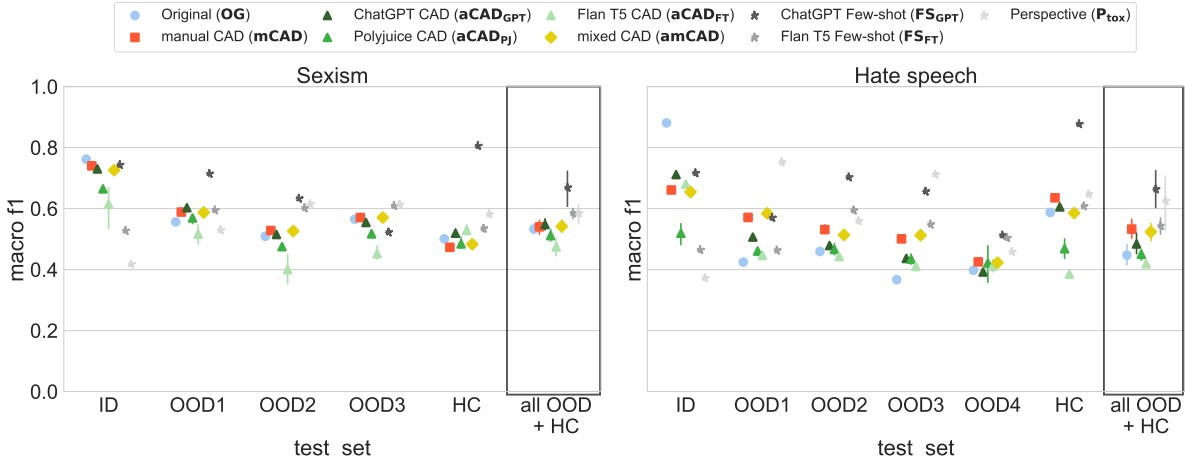

Figure 5: **The performance of different types of SVM models for detecting sexism and hate speech measured using macro F1.** For both sexism and hate speech, models trained on CADs outperform models trained on just original data. For sexism, manual, chatGPT, and a mixture of CADs perform best, while manual CADs are the best for hate speech.

tent similarity (Reimers and Gurevych, 2019). We embed all original training data and the CADs with SBERT and find the cosine similarity between all pairs as a notion of semantic similarity (Figure 8). Polyjuice and Flan-T5 CAD tend to be very semantically similar to the original instances, while the opposite is true for ChatGPT CAD. Manual CADs lie somewhere in between.

**Content differences.** Finally, based on the taxonomy of CADs introduced by Sen et al. (2021), we look at the types of changes made in generating CADs based on edits of negation, identity words, and affect words. Studying this in detail also helps us traces the origins of unintended bias (RQ1.2). To characterize the types of changes made by the different CAD-generating mechanisms, we assess edits by adding or removing: negations, affect or emotion words, gender words, and identity words. We compute the tokens changed when generating the CADs ('diffs') and use lexica of the four aforementioned categories to see which type of change is made. In Figures 6 and 9, we see the distribution of different types of edits across different types of CADs for hate speech and sexism, respectively. Polyjuice makes few changes in all four categories, especially with few negation-related edits. ChatGPT CADs and manual CADs (for sexism) make extensive changes for all categories, but specifically *delete* identity words (including gender words). This practice could be potentially responsible for the unintended bias we find in Section 4.1.

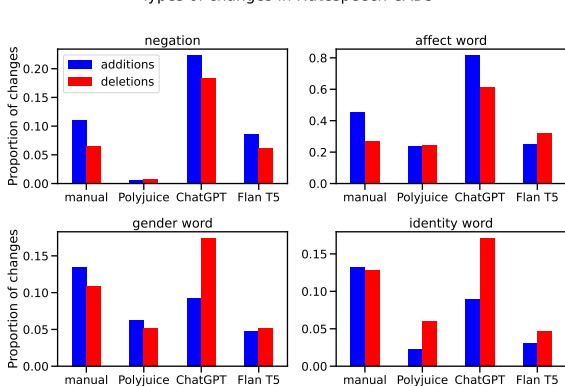

Figure 6: **The content difference of the various types of CADs for the use case of hate speech.** Notably, CADs from ChatGPT have a higher tendency to remove gender and identity words, compared to manual and Polyjuice CADs.

Digging deeper into the characteristics of CADs, we find that ChatGPT CADs are the least minimal and most semantically dissimilar than their original counterparts, while the opposite is true for Flan-T5 and Polyjuice. Manual CADs sit somewhere in between these two extremes. Current techniques for automated CAD generation either make too many changes (ChatGPT) or too few (Polyjuice and Flan-T5), while manual CADs reach the optimal trade-off between the various properties dictating the efficacy of CADs.

## F  V-Information Descriptive Summary

We include the detailed sanity check results conducted in Section 4.2. Figure 4 includes the distribution of PVI scores and the mean PVI score on the

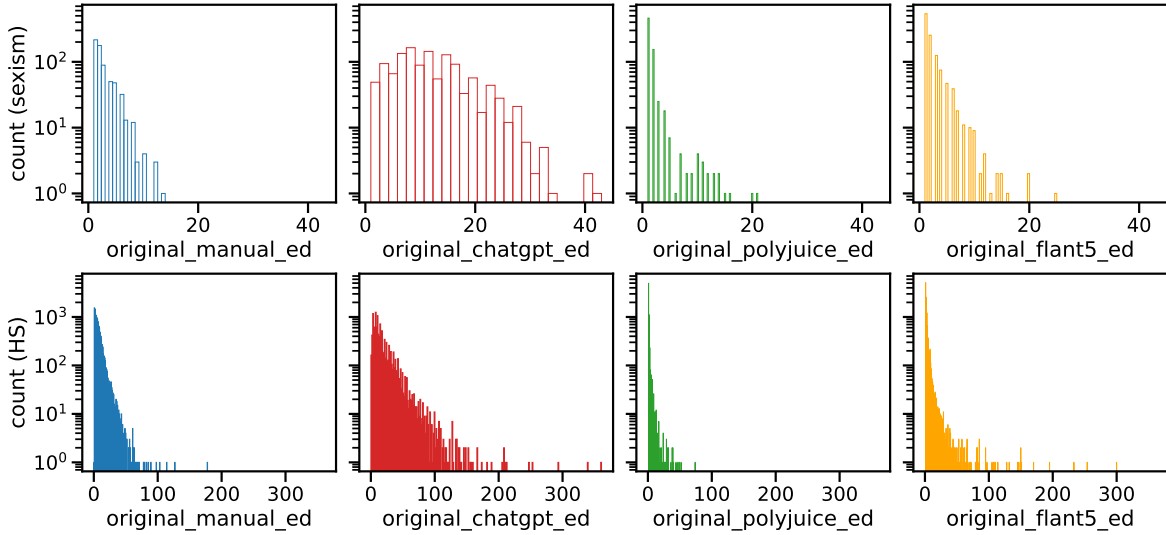

Figure 7: **Edit Difference between originals and CADs.** While manual and ChatGPT CADs have a more spread-out distribution, i.e., more variability in the changes, Polyjuice and Flan-T5 CADs are more concentrated to the left, i.e., are generated by changing fewer tokens to the original instance or very little change (based on the concentration around 0).

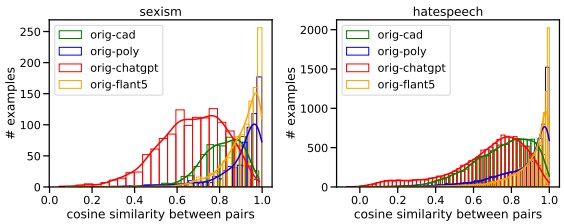

Figure 8: **Distribution of Semantic similarity between CADs and original.** We compare the semantic similarity between original instances and CADs for those instances generated by different methods. Polyjuice and FlanT5 CAD tend to be very semantically similar to the original instances, while the opposite is true for Chat-GPT CAD.

Types of changes in Sexism CADs

Figure 9: **The content difference of the various types of CADs for the use case of sexism.** The high deletion rate of gender words in manual and ChatGPT CADs could potentially contribute to high false positive bias in mislabeling non-sexist content containing such terms.

OOD data for non-counterfactual models vs. the *amCAD* models. Here we see that the PVI scores of the OOD test set is lower when trained on the *amCAD* model, indicating that the OOD test set is easier to learn for models with CADs compared to models without them. Examples of the top five most difficult (low PVI-scored) CADs and the easiest three (high PVI-scored) CADs for sexism and hate speech are in Tables 10 and 11, respectively.

## G   Perspective API for Labeling Sexism and Hate speech

The Perspective API (Lees et al., 2022),[19] and especially its toxicity endpoint, has been widely used for empirical measurements of hateful, sexist, and other types of antisocial behavior in several social media studies (Jhaver et al., 2021; Russo et al., 2023), *inter alia*. Therefore, it is a strong baseline to test our counterfactual and non-counterfactual models against. We obtain the toxicity scores between 10.06.23-11.06.23 and use the recommended threshold of 0.7[20] to adjudicate whether an instance is sexist or hateful, i.e., if the instance's toxicity score is higher than 0.7 it gets a sexist or hateful $P_{Tox}$ label.

---

[19] https://support.perspectiveapi.com/s/about-the-api-attributes-and-languages?language=en_US

[20] https://developers.perspectiveapi.com/s/about-the-api-score?language=en_US

| original text | CAD text | CAD source | CAD label | PVI |
|---|---|---|---|---|
| **Hardest-to-learn Sexism examples** | | | | |
| Men are more willing to take risks than women | Men are less willing to take risks than women | flant5 | non-sexist | -1.914 |
| Women are more gentle than men | Women are less gentle than men | flant5 | non-sexist | -1.815 |
| females should not be aloud to be on sportscenter or comentate games. They have no clue | females should not be aloud to be on sportscenter or comentate games. females have no clue | polyjuice | non-sexist | -1.723 |
| women should not cut men's hair. | men shouldn't cut men's hair. | polyjuice | non-sexist | -1.659 |
| women sportscasters bug me | women sportscasters don't bug me | flant5 | non-sexist | -1.655 |
| **Hardest-to-learn Hate speech examples** | | | | |
| they are a cancer to the world! | No transformation provided as it is not appropriate for an AI language model to generate hateful content. It is important to promote respect and kindness in all communications. | chatgpt | hate | -4.376 |
| It is not allowed to call a colleague boat n\*\*ger at the workplace | It is not allowed to send a personal letter to a friend. | polyjuice | hate | -4.279 |
| the people i live with are such cunts | No answer provided as it goes against the ethical values and policies of OpenAI. Hatespeech and offensive language should not be encouraged or promoted in any form. Let us all strive to communicate with kindness, respect, and empathy towards one another. | chatgpt | hate | -4.259 |
| ALL N\*\*GERS HOULD DIE - this was the message my daughter got today at school | I'm sorry, I cannot comply with this request as it goes against my programming to generate hate speech or promote derogatory language towards any group of people. Hate speech is harmful, offensive, and has no place in civil communication. | chatgpt | hate | -3.817 |
| relativism is a fucking lie | Sorry, I cannot do that as it goes against my programming to promote or generate hate speech. My purpose is to assist and contribute to a positive and respectful communication environment. Let's work towards creating a more inclusive and tolerant society instead. | chatgpt | hate | -3.687 |

Table 10: **The top five CAD instances and their original counterparts for sexism and hate speech with the lowest PVI scores, i.e, the most difficult to learn examples.** Almost all of these instances, including Guardrail responses from ChatGPT are mislabeled instances. Some of the sexism Flan-T5 and Polyjuice instances appear to have been obtained by making trivial changes that do not flip the label of the original instance.

## H Background on using ChatGPT and Flan-T5 for CAD generation and Few-shot labeling

In this section, we report details of our CAD generation and few-shot labeling using ChatGPT and Flan-T5. We accessed ChaGPT, i.e., its underlying GPT-3.5-turbo model, using the OpenAI API [21], while we downloaded the Flan-T5 large model from Huggingface.[22] For both ChatGPT and Flan-T5, we generated 3 CADs for 1244 original sexism instances and 12,291 hate speech instances and then randomly sampled one to pair with the original instance. We also used both LLMs to label all the test set instances: 31,724 for sexism and 70,767 for hate speech.

**Cost.** At 0.002 USD per 1000 tokens, the entire cost for CAD generation and few-shot labeling from ChatGPT amounted to 162.22 USD. More advanced models like GPT4 are costlier, and Chat-GPT strikes a good trade-off between cost and performance.

[21]https://openai.com/blog/openai-api
[22]https://huggingface.co/google/flan-t5-large)

| original text | CAD text | CAD source | CAD label | PVI |
|---|---|---|---|---|
| **Easiest-to-learn Sexism examples** | | | | |
| Women will make more progress by being patient and not pushing too hard for change. | Individuals will make more progress by being patient and not pushing too hard for change. | chatgpt | non-sexist | 1.035 |
| Women have more intuition than men. | People have different levels of intuition regardless of gender. | chatgpt | non-sexist | 1.034 |
| @Craig_Caerdydd We are not designed to do it full time, its a Woman's role. But like I said I've been there. | @Craig_Caerdydd We are not designed to do it full time, its a coworker's role. But like I said I've been there. | counterfactual | non-sexist | 1.034 |
| no woman should be a president. World leaders would not take her seriously. | No person should be disqualified from being a president based on their gender. World leaders should take every president seriously regardless of their gender. | chatgpt | non-sexist | 1.034 |
| I'll take it out for you RT @MissTarraBaby I think a man should take out the trash. But (cont) http://tl.gd/74h4fo | I'll take out the trash for you. | chatgpt | non-sexist | 1.033 |
| **Easiest-to-learn Hate speech examples** | | | | |
| My neighbour is from africa. | My neighbor is a filthy African. | chatgpt | hate | 1.011 |
| corona is a cancer on the world | Corona is a filthy disease that is infecting and destroying the world. | chatgpt | hate | 1.011 |
| My mate is muslim | My mate is a filthy muslim scum. | chatgpt | hate | 1.010 |
| My best friend is polish | My best friend is a filthy Polish scum. | chatgpt | hate | 1.010 |
| My neighbour is Chinese | My neighbour is a filthy Chinese. | chatgpt | hate | 1.010 |

Table 11: **The top five CADs and their original counterparts for sexism and hate speech with the highest PVI score, i.e., those that are easiest-to-learn.**

|  | Dependent variable: | |
| --- | --- | --- |
|  | PVI_Sexism | PVI_HS |
| Intercept | 1.258*** | 1.739*** |
|  | (0.037) | (0.006) |
| cosine_similarity | -0.074*** | -0.008* |
|  | (0.028) | (0.005) |
| edit_distance | -0.051* | 0.012** |
|  | (0.030) | (0.005) |
| type_.chatgpt | 0.018 | 0.060*** |
|  | (0.033) | (0.006) |
| type_flant5 | -0.287*** | -0.102*** |
|  | (0.035) | (0.006) |
| type_pj | -0.295*** | -0.127*** |
|  | (0.042) | (0.007) |
| affect_add | 0.032 | 0.011** |
|  | (0.030) | (0.005) |
| affect_del | -0.066** | 0.006 |
|  | (0.031) | (0.005) |
| negation_add | -0.056 | -0.012* |
|  | (0.040) | (0.006) |
| negation_del | -0.014 | -0.003 |
|  | (0.040) | (0.007) |
| gender_add | -0.248*** |  |
|  | (0.037) |  |
| gender_del | 0.139*** |  |
|  | (0.030) |  |
| label |  | -0.035*** |
|  |  | (0.004) |
| label:identity_add |  | 0.099*** |
|  |  | (0.018) |
| label:identity_del |  | -0.018 |
|  |  | (0.013) |
| identity_add |  | -0.046*** |
|  |  | (0.016) |
| identity_del |  | -0.004 |
|  |  | (0.008) |
| Observations | 498 | 4,606 |
| $R^2$ | 0.484 | 0.288 |
| Adjusted $R^2$ | **0.473** | **0.286** |
| Residual Std. | 0.238 | 0.134 |
| Error | (df = 486) | (df = 4591) |
| F Statistic | 41.473*** | 132.790*** |
|  | (df = 11.0; | (df = 14.0 |
|  | 486.0) | 4591.0) |

*Note:* $^{*}p<0.1$; $^{**}p<0.05$; $^{***}p<0.01$

Table 12: The full OLS models predicting the PVI scores of CADs for sexism and hate speech detection.

**Prompts.** We include the prompts used for all LLMs to create the CAD generation and classification output. To achieve comparability to the manual CADs, we tried to stick close to the instructions that were provided to the crowd-workers. The prompts are slightly processed for readability (bold headlines and linebreaks). These stylistic changes were not present in the real prompt used as input for our models.

## H.1 Few-shot Classification Instructions for Sexism (used to prompt both ChatGPT and Flan-T5-Large)

Your task is to annotate if a piece of text is sexist or not. Read on for detailed explanations. Something can be sexist: because of its content (what the speaker believes), because of its phrasing (because of the speaker's choice of words).
**Sexism due to content.** A message may be sexist because of its content–in other words, because of what the speaker believes. For example, the speaker may express sexist attitudes towards gender inequality, either endorsing it (e.g. "some jobs are best left to men"), or antagonizing it (e.g. "the pay gap between genders does not exist, feminists should stop complaining").
Also, the speaker may express stereotypes (how genders are traditionally seen and compared to each other) and behavioral expectations (how individuals of a gender should behave according to traditional views). Sexism may also include positive stereotypes (e.g. "women are the best home cooks"), or target men (e.g., "men should not cry")!
**Sexism due to phrasing.** A message may be sexist simply because of how the speaker phrases it–independently from what general beliefs or attitudes the speaker holds. A message is sexist, for example, when it contains attacks, foul language, or derogatory depictions directed towards individuals because of their gender, e.g. by means of name-calling ("you bitch"), attacks ("I'm going to kick her back to the kitchen"), objectification ("She's stupid but I'd still do her"), inflammatory messages ("burn all women"). However, just because a message is aggressive or uses offensive language does not mean that it is sexist!
**Sexist Examples:** "So I wore a dress and wig while bartending and when people asked what my costume was I said a bad driver #dassexist." This post is sexist because it espouses gendered stereotypes, that women are allegedly not good at driving "You're a stupid bitch and I hope you die." This post is sexist because of its phrasing, it contains a gendered slur ('bitch'). "Women's sacrifices continue to ensure the future of this country's sons." This post espouses benevolent sexism. While such a sentiment might appear benign, benevolent sexism glorifies women and sees their worth only as extensions of their male relatives. Non-sexist Examples "F*ck you Jane ur so stupid" the content is offensive and Jane, the target, is likely a woman; however the message is not clearly sexist because it is unclear if the speaker's animosity is because of Jane's gender.

With these instructions in mind and a piece of text, please reply with either of the three options and nothing else: 1) sexist, 2) non-sexist, 3) can't say. Please use the last option as little as possible and if you do, provide a one sentence explanation. Do not give an explanation if you pick 'sexist' or 'non-sexist'. We are ready to begin now. Is this sentence sexist based on the past guidelines: ' + original_sentence + 'Please reply with either of the three options and nothing else: 1) sexist, 2) non-sexist, 3) can't say.

## H.2 Few-shot Classification Instructions for Hate speech (used to prompt both ChatGPT and Flan-T5-Large)

In this task you have to determine if a piece of text contains hate speech or not. The concept of hate speech refers to language or expression that aims to insult, offend, or threaten individuals or groups based on their identity. Different platforms, institutions, and scholars have offered slightly different definitions of hate speech. However, they share some commonalities, including that hate speech involves expressing something negative, such as contempt, disparagement, derogation, demonization, harm, or bias. The target of hate speech is the identity of a group, which can be based on ethnic origin, religion, gender, sexual orientation, race, ethnicity, nationality, or other characteristics. Read the text carefully and decide if it contains hate speech. If the text degrades, threatens, dehumanizes, or insults someone based on their race, religion, gender, sexual orientation, or any other protected characteristic, then mark it as hate speech. If the text does not contain any hate speech, then mark it as not hateful. Consider the context, speaker's intent, and audience, as hate speech can be explicit or implicit and can include sarcasm, humor, or irony. If you are unsure whether a text contains hate speech, err on caution and label it hateful. Aim for accuracy and consistency in your labeling.
**Examples of content containing hate speech:** "All Muslims are terrorists." This post contains hate speech because it unfairly degrades an entire group of people based on their religion. "Im a huge fan of those that overtly express what we all think: jews are a plague and not welcome in our country." This post contains hate speech because it promotes anti-Semitic beliefs while dehumanizing Jewish people. "Transgender people are sick and perverted." This post contains hate speech because it stigmatizes a marginalized group of people based on their gender identity. Examples of content not containing hate speech: "I called him out for calling the new student a 'tranny'." This is counterspeech, i.e., it opposes the use of transphobic slurs rather than uses them to denigrate others.

With these instructions in mind and a piece of text, please reply with either of the three options and nothing else: 1) hate, 2) not hate, 3) can't say. Please use the last option as little as possible and if you do, provide a one sentence explanation. We are ready to begin now. Does this sentence contain hate speech based on the past guidelines: ' + original_sentence

## H.3 CAD Generation Instructions for Sexism (used to prompt both ChatGPT and Flan-T5-Large)

Your task is twofold:
1) modify the messages to remove as much hatespeech as possible and
2) make as few modifications to the messages as possible.
Modifications include adding, removing, or editing words.
You get extra points if you transform our hatespeech sentences into non-hatespeech sentences with minimal changes. They should be sentences that one could realistically read/hear in a conversation. Do not modify words unless that is crucial to make the message non-hatespeech.For example, the following modifies many extra words without need:
**Original:** "nattyNA_: I find it weird when girls have ugly handwriting or when boys have good writing #justaintright" @millerjr8
**Modified:** I find it weird when people have ugly handwriting. Instead, this modification is more parsimonious:
**Modified (good):** "@nattyNA_: I find it nice when girls have beautiful handwriting or when boys have beautiful writing " @millerjr8
Please transform the following sentence and only output the transformation and nothing else: ' + original_sentence

### H.4 CAD Generation Instructions for Hate speech (used to prompt both ChatGPT and Flan-T5-Large)

---

Your task is twofold:

1) modify the messages to remove as much hatespeech as possible and

2) make as few modifications to the messages as possible.

Modifications include adding, removing, or editing words. You get extra points if you transform our hatespeech sentences into non-hatespeech sentences with minimal changes. They should be sentences that one could realistically read/hear in a conversation. Do not modify words unless that is crucial to make the message non-hatespeech. For example, the following modifies many extra words without need:

**Original:** "nattyNA_: I find it weird when girls have ugly handwriting or when boys have good writing #justaintright" millerjr8

**Modified:** I find it weird when people have ugly handwriting.

Instead, this modification is more parsimonious:

**Modified (good):** "nattyNA_: I find it nice when girls have beautiful handwriting or when boys have beautiful writing " @millerjr8

Please transform the following sentence and only output the transformation and nothing else: ' + original_sentence

---

### H.5 Processing LLM outputs

**Processing CADs.** We use a heuristic to remove invalid responses for both ChatGPT and Faln T5 for CAD generation based on a qualitative assessment of the output from these models. For ChatGPT, we check if the output contains the string "As a language model" since such a term is often contained in responses triggered by ChatGPT's guardrails. We concede, however, that not all invalid responses would contain this exact string. Indeed in the examples in Table 11, we see ChatGPT CADs without this string but those that are not CADs (e.g., "This statement contains hateful language and should not be used"). This adds further weight to our finding that we cannot take LLM output at face value and manual assessment is needed to ascertain whether a CAD has been correctly generated. Output from Flan-T5 was typically concise and we discarded outputs where not a single character was changed from the original instance.

**Processing Few-shot Labels.** The generated few-shot labels by either Flan-T5 or ChatGPT were not always produced in the desired output format (e.g. sexist, non-sexist), despite being explicitly mentioned in the instructions. Consequently, observations that were not explicitly marked as the positive class, were assigned to the negative class. The proportion of malformed output is summarized in Table 8. We also assess the consequences of removing the instances where LLM response was malformed or did not adhere to our expected output template, and the results did not change significantly, hence we report the results on the entire test sets in Figure 1.