# OpenReview forum: "People Make Better Edits: Measuring the Efficacy of LLM-Generated Counterfactually Augmented Data for Harmful Language Detection"
_EMNLP/2023/Conference — EMNLP 2023 Main_

### Official Review · Reviewer_Hib4 · 2023-07-28

**Soundness:** 4

**Excitement:**

3: Ambivalent: It has merits (e.g., it reports state-of-the-art results, the idea is nice), but there are key weaknesses (e.g., it describes incremental work), and it can significantly benefit from another round of revision. However, I won't object to accepting it if my co-reviewers champion it.

**Missing References:**

The original RoBERTa citation (https://arxiv.org/abs/1907.11692) is missing
cite https://arxiv.org/abs/2107.00753

**Paper Topic And Main Contributions:**

Counterfactual data augmentation (CAD) is the process of augmenting a training dataset with minimally perturbed examples that flip the label [1]. The main reason to create this kind of data is to reduce the reliance of models on spurious correlations in the training data, thereby improving generalization.

Since creating this kind of data with humans is expensive, this paper studies the efficacy of automatically creating counterfactual data using a pretrained language models. The paper thoroughly analyzes several data creation strategies using ChatGPT, Flan-T5, PolyJuice. This data is then used to train a RoBERTa classifier. The paper experiments on datasets in harmful language detection, and measures both in-domain and out-domain generalization of the trained model.

Overall, the paper finds that human-written counterfactual data is the best, while ChatGPT-generated data is competitive. The authors thoroughly analyze their findings through information-theoretic dataset difficulty analysis, and find that FlanT5-generated data often does not flip the label, while ChatGPT-generated data is too easy to learn unlike human-written data.

[1] - https://arxiv.org/abs/1909.12434

**Reasons To Accept:**

1. The paper focuses on an important task in NLP --- harmful language detection. Reliance on spurious correlations in harmful language detection models can have serious social consequences, and it is important to study methods to reduce this reliance and improve generalization.

2. The paper presents a thorough experimental study analyzing several counterfactual data augmentation methods (ChatGPT, FlanT5, Polyjuice), as well as mixtures of automatic and manually created counterfactual data. Furthermore, the paper presents great analysis on dataset difficulty, and thoroughly investigates the reasoning behind the lack of generalization of automatic counterfactual data. The paper also includes some good baselines, including few-shot prompting with ChatGPT / FlanT5.


**Reasons To Reject:**


1. This paper would be stronger if experiments are conducted on stronger base model classifiers. The paper currently uses RoBERTa, which is a 2019 model. It would be interesting to see how these results generalize to newer or large language models, perhaps after fine-tuning the LLAMA models [1], or the FlanT5 models [2] themselves. This is important because it's likely that these base models are less dependent on spurious correlations in the first place (more in weakness #2), so the gap between automatic counterfactual data and manual counterfactual data will be lower. However, I appreciate the authors including ChatGPT few-shot prompting as a baseline, and agree that data contamination could explain its high performance on the task.

2. In my opinion, data augmentation has had a rollercoaster journey in NLP literature. Task-specific data augmentation methods like CAD were more popular pre-2020. However, after the introduction of large language models like GPT-3, they fell out of favor [5] due to the strong generalization capabilities of base models (related to weakness #1). Nevertheless, I think data augmentation is making a comeback in 2023, with a lot of great work on using automatically generated instruction tuning datasets [3, 4] to create general-purpose instruction following models. **This weakness is probably highly out of the scope of the current study**, but I am curious to know whether CAD techniques help with automatically created instruction tuning datasets --> in other words, minimal instruction changes that change the underlying task / labels. This would make the study more relevant to the current paradigms.

3. There are several commerical restrictions on using ChatGPT/GPT4 outputs, [6] says "one may not use output from the Services to develop models that compete with OpenAI". This may restrict the applicability of the proposed approach for downstream applications, and should be discussed further.

[1] - https://arxiv.org/abs/2302.13971
[2] - https://huggingface.co/docs/transformers/model_doc/flan-t5
[3] - https://arxiv.org/abs/2212.10560
[4] - https://arxiv.org/abs/2306.04751
[5] - https://twitter.com/_jasonwei/status/1526589109422026752
[6] - https://openai.com/policies/terms-of-use

**Reproducibility:**

4: Could mostly reproduce the results, but there may be some variation because of sample variance or minor variations in their interpretation of the protocol or method.

**Reviewer Confidence:**

3: Pretty sure, but there's a chance I missed something. Although I have a good feel for this area in general, I did not carefully check the paper's details, e.g., the math, experimental design, or novelty.

**Typos Grammar Style And Presentation Improvements:**

Fig 1 hard to read. Perhaps with the extra 9th page, it can be made bigger vertically, and the legend can use full names instead of shortforms? The legend could also be moved outside the plot.

It will be helpful to space out the paragraphs a bit more, shorten them and add more topical sentences per paragraph using the extra 9th page. This will help with the paper's readability.

---

> ### Author Rebuttal · Authors · 2023-08-28
>
> We are grateful to reviewer Hib4 for their positive feedback, helpful comments, and valuable literature suggestions. We will incorporate the latter two into the next version of our paper, including the constructive feedback on improving the presentation of the paper. We have the following responses to their critique:
>
> **Stronger Base Model Classifiers.** This is a good point, but we have some computational restrictions as do the typical computational social scientists who increasingly use BERT/RoBERTa classifiers in the realm of deep learning models. As the reviewer notes we do use the newer generation of LLMs like ChatGPT and FlanT5 for CAD generation as well as few-shot learning. For the few-shot Flan T5, we still note that RoBERTa models outperform them.
>
> We repeated these experiments with finetuned Flan T5 (Flan T5-base because that is the model we could use with our current hardware and time constraints for the rebuttal). We finetuned Flan-T5 by treating the classification task as a sequence-to-sequence problem, utilizing the same model parameter settings (learning rate) as in the original Flan T5 paper [1], also over 5 runs.
>
> We find similar performance as RoBERTa, with the finetuned Flan T5 model boosting the performance of models trained on Flan T5 and Polyjuice CADs. *However, the main tendencies seen in RoBERTa still hold, i.e, manual CADs are the best for hate speech, while a mix of manual and automated CADs is the best for sexism. Therefore, the results in our paper are not attributed to a particular model architecture.* We include the finetuned Flan T5 results for the combined OOD datasets, i.e., all OOD and Hatecheck in Table F of the rebuttal. We will include these results in the appendix of our paper.
>
> | construct   | mode         | Finetuned RoBERTa | Finetuned Flan T5 |
> |-------------|--------------|-------------------|-------------------|
> | hate speech | $aCAD_{FT}$  | 0.42 ± 0.066      | 0.563 ± 0.066     |
> |             | $aCAD_{GPT}$ | 0.609 ± 0.091     | 0.605 ± 0.059     |
> |             | $aCAD_{PJ}$  | 0.409 ± 0.087     | 0.524 ± 0.115     |
> |             | $OG$         | 0.532 ± 0.05      | 0.624 ± 0.057     |
> |             | $amCAD$      | 0.605 ± 0.084     | 0.585 ± 0.118     |
> |             | $mCAD$       | **0.673 ± 0.111** | **0.672 ± 0.086** |
> | sexism      | $aCAD_{FT}$  | 0.437 ± 0.099     | 0.561 ± 0.089     |
> |             | $aCAD_{GPT}$ | 0.595 ± 0.04      | 0.604 ± 0.024     |
> |             | $aCAD_{PJ}$  | 0.505 ± 0.119     | 0.565 ± 0.076     |
> |             | $OG$         | 0.591 ± 0.033     | 0.593 ± 0.025     |
> |             | $amCAD$      | **0.619 ± 0.03**  | **0.61 ± 0.046**  |
> |             | $mCAD$       | 0.596 ± 0.043     | 0.601 ± 0.032     |
>
> Table F: results of Finetuned RoBERTa vs. Finetuned Flan T5
>
> **whether CAD techniques help with automatically created instruction tuning dataset.** This is an intriguing suggestion and we would be very interested in working on this in the future. Our current work, however, shows that despite advances in LLMs, there are still questions about their few-shot and zero-shot labeling capabilities (as the reviewer observes ChatGPT’s data contamination issue, and Flan T5’s lackluster performance). And there is still the issue of computational cost — deploying Flan T5 is not accessible for all. Nonetheless, in the future, we would like to see how counterfactual prompts can improve zero-shot/few-shot labeling by LLMs. We thank the reviewer for this idea and the accompanying literature.
>
> **Commercial restriction on ChatGPT outputs.** Thank you for raising this important point. Our work serves as a proof of concept rather than commercial model building with the primary goal of facilitating computational social science research. Second, we build specialized models for the specific tasks of sexism and hate speech detection, instead of a general purpose labeling system like ChatGPT. Finally, legal use of LLM output, especially for research purposes is still nascent without any clear guidelines from research bodies. With all of this in mind, we believe we do not violate OpenAI’s policies, however, it is an important point and we will acknowledge it in the paper.
>
> References
>
> [1] Hyung Won Chung, Le Hou, Shayne Longpre, Barret Zoph, Yi Tay, William Fedus, Eric Li, Xuezhi Wang, Mostafa Dehghani, Siddhartha Brahma, et al. 2022. Scaling instruction-finetuned language models. arXiv preprint arXiv:2210.11416.

---

### Official Review · Reviewer_PwG4 · 2023-08-04

**Typos Grammar Style And Presentation Improvements:** 351 - missing year in citation
408 - …
**Soundness:** 3

**Excitement:**

3: Ambivalent: It has merits (e.g., it reports state-of-the-art results, the idea is nice), but there are key weaknesses (e.g., it describes incremental work), and it can significantly benefit from another round of revision. However, I won't object to accepting it if my co-reviewers champion it.

**Missing References:**

-

**Paper Topic And Main Contributions:**

The paper presents a study on the automatic generation of counterfactual training data. The authors implement several different strategies for augmentation and compare: 1) their usefulness in downstream tasks and 2) their internal properties.

The main contributions are:

- implementation of several strategies for the automatic generation of training data using counterfactuals
- experiments on training NLP models on augmented data
- analysis on the usefulness and properties of the generated/augmented data

**Questions For The Authors:**

Did you look at strategies that fail to generate examples? Why are there null values on "all" instances?

Did you try overgenerating and/or training on large synthetic datasets and how does data size affect performance?

Are all models using the same 50% (75%) original data or samples are different?

Is there any analysis on whether original/augmented pairs are both in training? how does that affect performance?

Did you consider using dataset cartography instead of  v-information as it was designed to measure usefulness for training?

Did you measure how training on e.g. chatgpt data improves (or reduces) pvi on other augmentation strategies?

Did you consider running statistical significance tests and/or multiple runs with aggregated scores and variance?

**Reasons To Accept:**

It's an interesting and potentially impactful topic that can lead to improvement in the field.

The authors have attempted to evaluate the augmentation/generation approaches from different perspectives, aiming to give a more holistic picture of their quality.

**Reasons To Reject:**

Almost all parts of the experimental design are severely flawed. The authors provide no justification for their experimental and evaluation decisions. There is no explanation on why particular experiments are run while others are not. For example, it makes sense to compare training sets of the same size, however it also makes sense to compare the effect of data size when training - the main advantage of using automated methods is that they scale, so it is important to measure how much you can gain by mass generating. The PVI experiments have large potential for analysis by actively controlling the training data, which has been missed completely by only using the amCAD model.

Key implementation details are missing or are put in appendixes. All of the data about the augmentation strategies is in appendix. There is no clarification for the data sampling use: 1) do all of the models use the same set of original "training" examples or those are shuffled differently? 2) do you include original + augmented examples or just one of them? 3) how does the decision in 2 affect performance?

There is no human validation of the augmentation, even if the authors report there are likely many errors. While validating all examples can be expensive and time consuming, doing 100 examples for each strategy/task is feasible even if the authors do it themselves.

No statistical significance tests on the results.

I really like the idea of the paper and the general approach taken by the authors. However, the experimental design needs substantial rework and/or justification, experiments and results likely need to be re-run in controlled conditions. The authors attempted to do too much, which results in too much focus and not enough details. I'd suggest they remove the regression analysis completely and focus on improving the rest of their experimental design.

**Reproducibility:**

3: Could reproduce the results with some difficulty. The settings of parameters are underspecified or subjectively determined; the training/evaluation data are not widely available.

**Reviewer Confidence:**

4: Quite sure. I tried to check the important points carefully. It's unlikely, though conceivable, that I missed something that should affect my ratings.

---

> ### Author Rebuttal · Authors · 2023-08-28
>
> We thank reviewer PwG404 for their comments. We are glad that the reviewer likes the idea and approach of the paper.
>
> We first respond to their critique in ‘reasons to reject’ and then answer their questions.
>
> **Replies to comments in ‘reasons to reject’:**
>
> **Experiment Design.** We took several steps to maintain the integrity of the experimental design and ensure the robustness of our results:
> 1) All training sets are of equal size (lines 215-217)
> 2) All training sets are balanced (lines 215-217)
> 3) We have several fully held-out out-of-domain test sets (4 for sexism, and 5 for hate speech), ruling out any data contamination for the fine tuned models (Table 4)
> 4) We report results for two different model architectures — fine tuned RoBERTa (fig 1) and Linear SVM (fig 4), plus fine tuned Flan T5 in the rebuttal in response to reviewer Hib4
> 5) We have 5 runs of experiments where the training sets were shuffled and report the mean and variance to reduce inflating results because of one specific split (line 269)
> 6) Now in the rebuttal, we also report that our models’ performance is backed by statistical significance (using McNemar’s test in Table B), adding more certainty of the robustness of our findings
>
> These steps ensure that our work is comparable to previous work testing the efficacy of manual CAD.
>
> While trying a higher proportion of CADs to original instances in training is interesting, it would require a fundamentally different experimental setup where these models are no longer comparable to models trained on manual CADs (more details and results in answer to QuestionB). These further experiments on size are also tangential to the main claims we make in this paper, i.e., that given equal amounts of training data, manual CADs are still better than automated CADs. While we agree that these and the cross-method PVI analyses could be interesting (more details in answer to QuestionF), as R1 points out, the paper is already dense. We hope to explore the issue of higher volume of automated CADs and cross-method PVI scores as follow-up work.
>
> **Key Implementation Details.**
>
> 1) All models do not use the same original training samples because different automated CAD generation techniques have varying coverage on the original instances (lines 167-175). As we have 5 runs for each model type, we bypass any issues related to one idiosyncratic training split (expanded in answer to QuestionC)
> 2) Yes, we always include the original and augmented pair as stated in lines 124-125 and 161-163. This is inline with past work [1, 3].
> 3) When CADs are not paired in the training data, the models perform slightly worse. However, our results remain unchanged, i.e., a mixture of manual and automated CADs is best for sexism, while manual CADs are best for hate speech (see answer to QuestionD for details).
>
> Most of these points (except the last since it was not the main focus of our work) are in the main paper. However, due to space limitations, we had to move more detailed explanations to the appendix. We will move them to the main paper for a tentative final version.
>
> **Human Validation.** The reviewer mentions the lack of human validation of automated CADs’ labels as a reason to reject. However, we would like to point out that we have done a human validation and reported it in the paper to determine the extent of mislabeling, exactly as the reviewer suggests — 300 examples each for sexism and hate speech were annotated by two of the paper's authors (lines 507-522). Therefore, our conclusion that several automated CADs have errors, i.e., do not flip the label is backed by evidence.
>
> **Statistical Significance.** We do report statistical significance for RQ2 in the regression table (Table 5) — the main takeaways are backed by significant coefficients. For RQ1, given that we compare so many models, it would be infeasible to report significance tests for all comparisons. In the rebuttal, we now conduct the McNemar’s test [2] and find that the results of RQ1 are statistically significant (details in answer to QuestionG)
>
> We now reply to the reviewer’s questions in **‘Questions for the authors’**:
>
> **QuestionA. Did you look at strategies that fail to generate examples? Why are there null values on "all" instances?**
>
> We thank the reviewer for raising this question, which points to important issues on which original instances can be counterfactually perturbed and using which strategies (i.e., control codes). However, these questions are broad, so much so that they would grant an independent research line. In this paper, we gave a much more constrained explanation in lines 140-143. We did look into the Polyjuice strategy that returned null values for all. As we wrote, the ‘shuffle’ code returned null for all instances, probably because our training data consists of short tweets (see example in Table 1) which typically do not have multiple key phrases that can be shuffled around.
>
> **QuestionB. Did you try overgenerating and/or training on large synthetic datasets and how does data size affect performance?**
>
> Previous work [1, 3, 4] has shown that CAD proportions higher than 50% (or 25% if only one-sided CADs are present) can actually degrade OOD performance. We thus opted for the setup of keeping the proportions of CADs as half of the original data, in line with previous work.
>
> We have conducted some preliminary experiments on having more CADs than originals, however, we do not explore this in detail in this current paper due to the following reasons:
> - For sexism where there are no sexist CADs and having more CADs than originals would mean having an unbalanced training dataset, making it difficult to assess if a model’s success on an OOD test set is because of the CADs or the data imbalance.
> - We could do these experiments for just hate speech, but this would imply a different experimental setup that is incomparable to past work. Assessing the effect of varying data size is therefore not within the current scope of this work.
>
> Nonetheless, we include the preliminary results for higher proportions of ChatGPT CADs for hate speech in the rebuttal (Table A). We conduct these experiments for ChatGPT CADs only since they were the most effective among automated CADs. Our findings match previous work — *higher proportions of CADs hinder out-of-domain generalizability rather than facilitating it*, i.e., models trained on twice as many ChatGPT CADs ($aCAD^2_{GPT}$) as originals are worse than models trained on original-single ChatGPT CAD pairs ($aCAD_{GPT}$), despite more training data. Note that the data is still balanced by classes, but $aCAD^2_{GPT}$ models were trained on more data, i.e., twice as many CADs. Like experiments in RQ1, these experiments were done over 5 runs.
>
> However, we reiterate that assessing the correct proportion of CADs requires extensive experiments and is out of scope for this current work, where our main focus is comparing automated and manual CADs in a fair comparison.
>
> | model      | test_set   | HC                | ID                | OOD1              | OOD2              | OOD3              | OOD4              | all OOD (OOD + HC) |
> |------------|------------|-------------------|-------------------|-------------------|-------------------|-------------------|-------------------|--------------------|
> | RoBERTA    | $aCAD_{GPT}$   | **0.734 ± 0.02**  | **0.885 ± 0.009** | 0.678 ± 0.037     | **0.546 ± 0.029** | 0.582 ± 0.04      | **0.504 ± 0.022** | **0.609 ± 0.091**  |
> |            | $aCAD^2_{GPT}$ | 0.659 ± 0.037     | 0.857 ± 0.015     | **0.687 ± 0.043** | 0.533 ± 0.044     | **0.61 ± 0.025**  | 0.475 ± 0.079     | 0.593 ± 0.092      |
> | Linear SVM | $aCAD_{GPT}$   | **0.604 ± 0.009** | **0.714 ± 0.008** | **0.509 ± 0.011** | **0.482 ± 0.012** | **0.438 ± 0.009** | **0.397 ± 0.015** | **0.486 ± 0.072**  |
> |            | $aCAD^2_{GPT}$ | 0.587 ± 0.011     | 0.608 ± 0.004     | 0.517 ± 0.012     | 0.452 ± 0.008     | 0.434 ± 0.014     | 0.387 ± 0.006     | 0.475 ± 0.072      |
>
> Table A. Comparing models trained on twice as many originals ($aCAD^2_{GPT}$) vs. models trained on single CAD-original pairs ($aCAD_{GPT}$). In line with past work [1, 3, 4] more CADs reduce overall performance, indicating that simply adding extra CADs via automation is not the best path forward.
>
> **QuestionC. Are all models using the same 50% (75%) original data or samples are different?**
>
> As mentioned in answer to QuestionG, we have reported results based on 5 runs (line 269 in the paper). This means that for each mode, we have a balanced training dataset that is obtained by randomly sampling data based on Table 2. Since we balance the datasets, we downsample one of the classes of the original dataset. Therefore the training sets of all methods are drawn from the same source for each individual mode but are different because they have CADs from different sources — $mCAD$ has manual CADs and $aCAD_{GPT}$ has GPT CADs, so the training set cannot be the same for all models. Moreover, since different CAD generating techniques have varying coverage (lines 167-175), we cannot have the same original examples for all models in training set for one particular run. To ensure that our results are not an artifact of a surprisingly beneficial random split, we run all experiments over 5 runs.
>
> **QuestionD. Is there any analysis on whether original/augmented pairs are both in training? how does that affect performance?**
>
> Both original and augmented were paired together in the training set as stated in lines 124-125 and 161-163, as done in past research [1, 3]. We include results of unpaired CADs on the combined OOD datasets below (Table B) and see that these models tend to perform similarly or slightly worse than paired CADs. However, our results w.r.t. performance of automated vs. manual CADs still holds. This would indicate that full pairing is more advantageous, however, future work on this is needed.
>
> | construct   | mode         | paired            | unpaired          |
> |-------------|--------------|-------------------|-------------------|
> | hate speech | $aCAD_{FT}$  | 0.42 ± 0.066      | 0.463 ± 0.103     |
> |             | $aCAD_{GPT}$ | 0.609 ± 0.091     | 0.602 ± 0.09      |
> |             | $aCAD_{PJ}$  | 0.409 ± 0.087     | 0.457 ± 0.056     |
> |             | $amCAD$      | 0.605 ± 0.084     | 0.546 ± 0.078     |
> |             | $mCAD$       | **0.673 ± 0.111** | **0.625 ± 0.074** |
> | sexism      | $aCAD_{FT}$  | 0.437 ± 0.099     | 0.522 ± 0.076     |
> |             | $aCAD_{GPT}$ | 0.595 ± 0.04      | 0.587 ± 0.086     |
> |             | $aCAD_{PJ}$  | 0.505 ± 0.119     | 0.512 ± 0.081     |
> |             | $amCAD$      | **0.619 ± 0.03**  | **0.589 ± 0.08**  |
> |             | $mCAD$       | 0.596 ± 0.043     | 0.585 ± 0.089     |
>
> Table B: Models trained on paired CADs vs. unpaired CADs.
>
> **QuestionE. Did you consider using dataset cartography instead of v-information as it was designed to measure usefulness for training?**
>
> We thank the reviewer for bringing up this point. V-information and dataset cartography are related (Appendix of Ethayaraj et al., 2022 [5]); it would be partially redundant to use both; Dataset Cartography has two metrics (variability and confidence) instead of one in V-information (Pointwise V-Information or PVI), making PVI scores more suitable for interpreting the properties of data instances. Moreover, V-information also provides aggregate measures of dataset difficulty which we report in Figure 2, which we could not do with dataset cartography. In sum, V-information is more concise and easier to integrate with the many variables and analyses we have in this current paper, and it is more theoretically aligned with the goals of the paper.
>
> **QuestionF. Did you measure how training on e.g. chatgpt data improves (or reduces) pvi on other augmentation strategies?**
>
> The goal of this paper was not to look at whether training on certain types of CADs makes other CADs easier to learn, but to assess the efficacy of automated CADs in general, especially when compared to manual CADs. The analysis the reviewer suggested would tell us if training on ChatGPT CADs makes learning other types of CADs easier — it wouldn’t tell us if automated CADs are easier or harder to learn than manual CADs. The reason for using the $amCAD$ model in the V-information experiments was that we can now compare the difficulty of both automated and manual CADs (lines 254-256). While the suggested experiment is interesting and we can look into it for future research, it would be out of scope for the current paper — indeed a systematic cross-method evaluation would be a full paper in and of itself. However, the current paper and the analysis of general automated CAD quality is needed to lay the foundation for doing some of the further experiments the reviewer suggests, i.e., we first need to know how they compare against the state-of-the-art i.e., manual CADs, and what their shortcomings are.
>
> **QuestionG. Did you consider running statistical significance tests and/or multiple runs with aggregated scores and variance?**
>
> **Significance Tests.** Given that we compare so many models, we did not report significance tests for all comparisons, however, for the main takeaways (1. $mCAD$ is better than $OG$ for both sexism and hate speech, 2. $mCAD$ is better than $aCAD_{GPT}$ for hate speech, 3. $amCAD$ is better than $aCAD_{GPT}$ for sexism) using the McNemar test [2] our findings are significant, except in one single run for sexism. We report these results in Table B. We will also add this to the appendix of our paper. Due to space constraints, we only report the results on the combined OOD datasets, but the results were also significant for the individual OOD datasets.
>
> | construct  | run | dataset                | mcnemar         | mcnemar P value  | model1    | model2       |
> |------------|-----|------------------------|-----------------|------------------|-----------|--------------|
> | sexism     | 0   | All OOD (OOD + HC)     | 55.18581081     | 1.10E-13         | $OG$        | $mCAD$         |
> |            | 0   | All OOD (OOD + HC)     | 359.5694618     | 3.49E-80         | $mCAD$      | $aCAD_{GPT}$     |
> |            | 0   | All OOD (OOD + HC)     | 126.1576971     | 2.84E-29         | $amCAD$     | $aCAD_{GPT}$     |
> |            | 1   | All OOD (OOD + HC)     | 953.8163619     | 1.96E-209        | $OG$        | $mCAD$         |
> |            | 1   | All OOD (OOD + HC)     | 122.2522026     | 2.03E-28         | $mCAD$      | $aCAD_{GPT}$     |
> |            | 1   | All OOD (OOD + HC)     | 99.04454769     | 2.47E-23         | $amCAD$     | $aCAD_{GPT}$     |
> |            | 2   | All OOD (OOD + HC)     | 1376.569069     | 2.59E-301        | $OG$        | $mCAD$         |
> |            | 2   | All OOD (OOD + HC)     | 842.2262547     | 3.56E-185        | $mCAD$      | $aCAD_{GPT}$     |
> |            | 2   | *All OOD (OOD + HC)* | *1.691980225* | *0.1933401375* | *amCAD* | *aCAD_GPT* |
> |            | 3   | All OOD (OOD + HC)     | 4279.530119     | 0                | $OG$        | $mCAD$         |
> |            | 3   | All OOD (OOD + HC)     | 1311.523288     | 3.54E-287        | $mCAD$      | $aCAD_{GPT}$     |
> |            | 3   | All OOD (OOD + HC)     | 116.2846413     | 4.12E-27         | $amCAD$     | $aCAD_{GPT}$     |
> |            | 4   | All OOD (OOD + HC)     | 1592.793376     | 0                | $OG$        | $mCAD$         |
> |            | 4   | All OOD (OOD + HC)     | 3454.453729     | 0                | $mCAD$      | $aCAD_{GPT}$     |
> |            | 4   | All OOD (OOD + HC)     | 171.6184932     | 3.28E-39         | $amCAD$     | $aCAD_{GPT}$     |
> | hatespeech | 0   | All OOD (OOD + HC)     | 954.1794424     | 1.64E-209        | $OG$        | $mCAD$         |
> |            | 0   | All OOD (OOD + HC)     | 11317.37678     | 0                | $mCAD$      | $aCAD_{GPT}$     |
> |            | 0   | All OOD (OOD + HC)     | 305.6896526     | 1.90E-68         | $amCAD$     | $aCAD_{GPT}$     |
> |            | 1   | All OOD (OOD + HC)     | 2791.831546     | 0                | $OG$        | $mCAD$         |
> |            | 1   | All OOD (OOD + HC)     | 8512.32061      | 0                | $mCAD$      | $aCAD_{GPT}$     |
> |            | 1   | All OOD (OOD + HC)     | 2207.605596     | 0                | $amCAD$     | $aCAD_{GPT}$     |
> |            | 2   | All OOD (OOD + HC)     | 345.4147134     | 4.22E-77         | $OG$        | $mCAD$         |
> |            | 2   | All OOD (OOD + HC)     | 13411.88159     | 0                | $mCAD$      | $aCAD_{GPT}$     |
> |            | 2   | All OOD (OOD + HC)     | 712.4484629     | 5.87E-157        | $amCAD$     | $aCAD_{GPT}$     |
> |            | 3   | All OOD (OOD + HC)     | 4623.373874     | 0                | $OG$        | $mCAD$         |
> |            | 3   | All OOD (OOD + HC)     | 3278.335217     | 0                | $mCAD$      | $aCAD_{GPT}$     |
> |            | 3   | All OOD (OOD + HC)     | 1573.693207     | 0                | $amCAD$     | $aCAD_{GPT}$     |
> |            | 4   | All OOD (OOD + HC)     | 559.0799142     | 1.33E-123        | $OG$        | $mCAD$         |
> |            | 4   | All OOD (OOD + HC)     | 17744.84317     | 0                | $mCAD$      | $aCAD_{GPT}$     |
> |            | 4   | All OOD (OOD + HC)     | 1944.128811     | 0                | $amCAD$     | $aCAD_{GPT}$     |
>
> Table C: Significance tests between 1) $OG$ and $mCAD$ models, 2) $mCAD$ and $aCAD_{GPT}$, and 3) $amCAD$ and $aCAD_{GPT}$ models using McNemar's significance test (RoBERTa models).
>
> **Multiple runs.** We do have multiple runs (5) and report the mean and variance in Figure 1 and 4, as stated in line 269 in the main paper and expanded in the Appendix (lines 1048-1050). We have provided a tabular version of the results in the rebuttal (Table C for RoBERTa and Table D for Linear SVM) if that is easier to interpret. Though given that we compare 9 methods across 5-6 test sets for two NLP tasks, interpreting the table is also not trivial, while the table takes up more space in the paper. We will include it in the appendix of the paper for reference
>
> | construct  | mode         | HC            | ID            | OOD1          | OOD2          | OOD3          | OOD4          | all OOD (OOD + HC) |
> |------------|--------------|---------------|---------------|---------------|---------------|---------------|---------------|--------------------|
> | hatespeech | $FS_{FT}$    | 0.608 ± 0.002 | 0.464 ± 0.004 | 0.463 ± 0.002 | 0.594 ± 0.004 | 0.548 ± 0.002 | 0.504 ± 0.0   | 0.543 ± 0.056      |
> |            | $FS_{GPT}$   | 0.877 ± 0.001 | 0.716 ± 0.002 | 0.57 ± 0.002  | 0.703 ± 0.002 | 0.656 ± 0.0   | 0.512 ± 0.002 | 0.663 ± 0.13       |
> |            | $P_{tox}$    | 0.647 ± 0.0   | 0.372 ± 0.0   | 0.752 ± 0.0   | 0.559 ± 0.0   | 0.712 ± 0.0   | 0.458 ± 0.0   | 0.625 ± 0.119      |
> |            | $aCAD_{FT}$  | 0.343 ± 0.055 | 0.486 ± 0.055 | 0.456 ± 0.036 | 0.457 ± 0.027 | 0.444 ± 0.051 | 0.401 ± 0.08  | 0.42 ± 0.066       |
> |            | $aCAD_{GPT}$ | 0.734 ± 0.02  | 0.885 ± 0.009 | 0.678 ± 0.037 | 0.546 ± 0.029 | 0.582 ± 0.04  | 0.504 ± 0.022 | 0.609 ± 0.091      |
> |            | $aCAD_{PJ}$  | 0.367 ± 0.135 | 0.503 ± 0.157 | 0.416 ± 0.074 | 0.441 ± 0.067 | 0.399 ± 0.1   | 0.424 ± 0.059 | 0.409 ± 0.087      |
> |            | $OG$           | 0.586 ± 0.054 | 0.967 ± 0.005 | 0.571 ± 0.03  | 0.502 ± 0.031 | 0.487 ± 0.028 | 0.512 ± 0.004 | 0.532 ± 0.05       |
> |            | $amCAD$        | 0.671 ± 0.069 | 0.791 ± 0.014 | 0.699 ± 0.032 | 0.557 ± 0.02  | 0.602 ± 0.04  | 0.495 ± 0.013 | 0.605 ± 0.084      |
> |            | $mCAD$         | 0.832 ± 0.021 | 0.881 ± 0.01  | 0.757 ± 0.007 | 0.622 ± 0.017 | 0.628 ± 0.008 | 0.525 ± 0.007 | **0.673 ± 0.111**  |
> | sexism     | $FS_{FT}$    | 0.534 ± 0.01  | 0.527 ± 0.003 | 0.594 ± 0.002 | 0.601 ± 0.002 | 0.609 ± 0.002 | N/A           | 0.585 ± 0.031      |
> |            | $FS_{GPT}$   | 0.805 ± 0.004 | 0.743 ± 0.001 | 0.714 ± 0.002 | 0.633 ± 0.002 | 0.522 ± 0.001 |               | 0.669 ± 0.109      |
> |            | $P_{tox}$    | 0.581 ± 0.0   | 0.417 ± 0.0   | 0.529 ± 0.0   | 0.615 ± 0.0   | 0.613 ± 0.0   |               | 0.584 ± 0.04       |
> |            | $aCAD_{FT}$  | 0.435 ± 0.089 | 0.703 ± 0.034 | 0.532 ± 0.052 | 0.36 ± 0.097  | 0.422 ± 0.086 |               | 0.437 ± 0.099      |
> |            | $aCAD_{GPT}$ | 0.565 ± 0.067 | 0.782 ± 0.009 | 0.614 ± 0.023 | 0.588 ± 0.024 | 0.611 ± 0.013 |               | 0.595 ± 0.04       |
> |            | $aCAD_{PJ}$  | 0.349 ± 0.06  | 0.76 ± 0.047  | 0.568 ± 0.051 | 0.516 ± 0.107 | 0.586 ± 0.077 |               | 0.505 ± 0.119      |
> |            | $OG$           | 0.547 ± 0.018 | 0.81 ± 0.004  | 0.608 ± 0.03  | 0.588 ± 0.009 | 0.621 ± 0.005 |               | 0.591 ± 0.033      |
> |            | $amCAD$        | 0.587 ± 0.029 | 0.78 ± 0.008  | 0.642 ± 0.022 | 0.61 ± 0.019  | 0.636 ± 0.011 |               | **0.619 ± 0.03**   |
> |            | $mCAD$         | 0.544 ± 0.024 | 0.761 ± 0.007 | 0.655 ± 0.014 | 0.587 ± 0.01  | 0.6 ± 0.007   |               | 0.596 ± 0.043      |
>
> Table D: RoBERTa results (macro F1 over 5 runs for each test sets)
>
> | construct  | mode         | HC            | ID            | OOD1          | OOD2          | OOD3          | OOD4          | all OOD (OOD + HC) |
> |------------|--------------|---------------|---------------|---------------|---------------|---------------|---------------|--------------------|
> | hatespeech | $aCAD_{FT}$  | 0.384 ± 0.009 | 0.675 ± 0.013 | 0.449 ± 0.016 | 0.444 ± 0.009 | 0.415 ± 0.017 | 0.405 ± 0.025 | 0.419 ± 0.029      |
> | hatespeech | $aCAD_{GPT}$ | 0.604 ± 0.009 | 0.714 ± 0.008 | 0.509 ± 0.011 | 0.482 ± 0.012 | 0.438 ± 0.009 | 0.397 ± 0.015 | 0.486 ± 0.072      |
> | hatespeech | $aCAD_{PJ}$  | 0.471 ± 0.033 | 0.516 ± 0.035 | 0.454 ± 0.021 | 0.461 ± 0.022 | 0.434 ± 0.018 | 0.431 ± 0.065 | 0.45 ± 0.037       |
> | hatespeech | $OG$           | 0.588 ± 0.003 | 0.881 ± 0.002 | 0.424 ± 0.003 | 0.46 ± 0.002  | 0.366 ± 0.003 | 0.401 ± 0.007 | 0.448 ± 0.078      |
> | hatespeech | $amCAD$        | 0.584 ± 0.006 | 0.658 ± 0.014 | 0.577 ± 0.017 | 0.516 ± 0.008 | 0.507 ± 0.018 | 0.423 ± 0.016 | 0.521 ± 0.061      |
> | hatespeech | $mCAD$         | 0.635 ± 0.013 | 0.667 ± 0.013 | 0.572 ± 0.004 | 0.532 ± 0.015 | 0.501 ± 0.003 | 0.425 ± 0.009 | **0.533 ± 0.072**  |
> | sexism     | $aCAD_{FT}$  | 0.517 ± 0.032 | 0.621 ± 0.07  | 0.522 ± 0.036 | 0.416 ± 0.061 | 0.464 ± 0.037 | N/A     | 0.48 ± 0.059       |
> | sexism     | $aCAD_{GPT}$ | 0.516 ± 0.017 | 0.729 ± 0.005 | 0.602 ± 0.003 | 0.516 ± 0.009 | 0.554 ± 0.007 |               | **0.547 ± 0.037**  |
> | sexism     | $aCAD_{PJ}$  | 0.492 ± 0.025 | 0.665 ± 0.006 | 0.572 ± 0.018 | 0.469 ± 0.016 | 0.513 ± 0.016 |               | 0.511 ± 0.043      |
> | sexism     | $OG$           | 0.506 ± 0.018 | 0.762 ± 0.004 | 0.555 ± 0.004 | 0.507 ± 0.006 | 0.565 ± 0.004 |               | 0.533 ± 0.029      |
> | sexism     | $amCAD$        | 0.487 ± 0.015 | 0.725 ± 0.006 | 0.588 ± 0.006 | 0.524 ± 0.007 | 0.57 ± 0.003  |               | 0.542 ± 0.042      |
> | sexism     | $mCAD$         | 0.477 ± 0.011 | 0.739 ± 0.005 | 0.589 ± 0.01  | 0.528 ± 0.004 | 0.572 ± 0.004 |               | 0.541 ± 0.045      |
>
> Table E: Linear SVM results (macro F1 over 5 runs for each test sets)
>
> References
>
> [1] Kaushik, Divyansh, Eduard Hovy, and Zachary Lipton. "Learning The Difference That Makes A Difference With Counterfactually-Augmented Data." International Conference on Learning Representations. 2019.
>
> [2] Dietterich, Thomas G. "Approximate statistical tests for comparing supervised classification learning algorithms." Neural computation 10.7 (1998): 1895-1923.
>
> [3] Sen, Indira, et al. "How Does Counterfactually Augmented Data Impact Models for Social Computing Constructs?." Proceedings of the 2021 Conference on Empirical Methods in Natural Language Processing. 2021.
>
> [4] Wu, T., et al. "Polyjuice: Generating Counterfactuals for Explaining, Evaluating, and Improving Models." Joint Conference of the 59th Annual Meeting of the Association for Computational Linguistics and the 11th International Joint Conference on Natural Language Processing (ACL-IJCNLP 2021). 2021.
>
> [5] Ethayarajh, Kawin, Yejin Choi, and Swabha Swayamdipta. "Understanding Dataset Difficulty with \mathcalV -Usable Information." International Conference on Machine Learning. PMLR, 2022.

---

### Official Review · Reviewer_z9F6 · 2023-08-05

**Soundness:** 4

**Excitement:**

3: Ambivalent: It has merits (e.g., it reports state-of-the-art results, the idea is nice), but there are key weaknesses (e.g., it describes incremental work), and it can significantly benefit from another round of revision. However, I won't object to accepting it if my co-reviewers champion it.

**Paper Topic And Main Contributions:**

This paper explores the usefulness of automatically generated Counterfactually Augmented Data (CADs) points. These CADs are generated using three generative models: Polyjuice, ChatGPT, and FlanT5. The original CADs were manually generated in previous works. The generative models were used to alter one word in each of the CADs creating a set of counterfactuals.

Once the authors automatically generate CADs, they fine-tune RoBERTa models and SVM models with Fasttext embeddings on the data sets to perform two types of classification: hate speech (binary hateful vs non-hateful), and sexism (binary sexist vs not sexist). They also compare the results of their models to a selection of baselines. These classification tasks are used to evaluate the usefulness of automatically generated CADs in improving model robustness compared to manually-generated CADs. The models the authors present are tested on various out-of-domain datasets.

The authors show that none of the automatically generated CADs beat the manually generated ones. However, the CADs generated by ChatGPT were the best scoring among the automatically generated CADs. Nonetheless, they also show that automatically generated CADs improve out-of-domain generalizability of both sexism and hate speech
detection models. The authors also draw the attention towards the necessity of manual checking, and that current LLMs are should not be trusted and used for fully automated CAD generation. One additional drawback of using these types of LLMs to generate counterfactuals is that they tend to introduce changes that are not sufficient to flip the original labels.

The paper is clear and well written. The ideas and argumentation are clearly stated and described. The results are well reported and clearly analysed. However, I am not sure about the novelty of the work.


**Reasons To Accept:**

- Nice contribution to the discussions around automatically vs manually generated counterfactual datasets.
- In depth analysis of the results.
- Broad experimental setup, with detailed descriptions.
- Well written paper.

**Reasons To Reject:**

- I cannot see any. The only negative thing I could think of is that the paper is a times a bit dense to read. But a lot of work has been done and needs to be reported, I therefore do not see this as a weakness.

**Reproducibility:**

3: Could reproduce the results with some difficulty. The settings of parameters are underspecified or subjectively determined; the training/evaluation data are not widely available.

**Reviewer Confidence:**

3: Pretty sure, but there's a chance I missed something. Although I have a good feel for this area in general, I did not carefully check the paper's details, e.g., the math, experimental design, or novelty.

---

> ### Author Rebuttal · Authors · 2023-08-28
>
> We thank reviewer z9F6 for their positive and constructive feedback. We appreciate that they found our analysis to be thorough and detailed. We concede that the paper is indeed dense, but with an extra page, we can extrapolate on some of the points and make it more readable.
>
> In terms of novelty, while past work has used LLMs like GPT2 for generating CADs [Polyjuice], as far as we are aware, no other paper has:
> 1) investigated the direct comparison of manually generated CADs and automated CADs,
> 2) investigated the specific shortcomings of automated CADs
> 3) used recent instruction-based LLMs’ efficacy in CAD generation
> 4) finally for the use cases of harmful language detection
>
> Wu et al. compare manual and automated CADs for Natural Language Inference [1] and find what makes automated CADs effective, while Howard et al. do so for sentiment analysis [2] while discussing that future work should expand to other tasks. Therefore, similar efforts are needed for sexism and hate speech detection. Therefore, our work is the first step to show what the potentials and pitfalls of these CADs are and to surface which of their weaknesses future work needs attention. We will modify the next version of our paper to highlight this novelty more clearly.
>
> References
>
> [1] Wu, T., et al. "Polyjuice: Generating Counterfactuals for Explaining, Evaluating, and Improving Models." Joint Conference of the 59th Annual Meeting of the Association for Computational Linguistics and the 11th International Joint Conference on Natural Language Processing (ACL-IJCNLP 2021). 2021.
>
> [2] Howard, Phillip, et al. "NeuroCounterfactuals: Beyond Minimal-Edit Counterfactuals for Richer Data Augmentation." Findings of the Association for Computational Linguistics: EMNLP 2022. 2022.

---

### Meta-Review · Area_Chair_HzW4 · 2023-09-19

**Recommendation:** 4

**Metareview:**

This work explores various ways to generate Counterfactually Augmented Data creation, such as through ChatGPT, FLAN-T5 and PolyJuice. Authors show that aside from manual CADs, ChatGPT is also an effective tool for such data augmentation. Author has primarily used Finetuned RoBERTa classifier and the task of harmful language detection to showcase the efficacy. Reviewers agree that the discussion and comparison around manual vs automated data augmentation is an interesting contribution. Most of the points such as manual evaluation of the generated data, concentrating only on RoBERTa has been addressed by the authors during rebuttal period. It is interesting work and the authors should incorporate the suggestions made by the reviewers to the best of their capability.

---

### Decision · Program_Chairs · 2023-10-07

**Decision:**

Accept-Main

**Comment:**

This work explores various ways to generate Counterfactually Augmented Data creation, such as through ChatGPT, FLAN-T5 and PolyJuice. Authors show that aside from manual CADs, ChatGPT is also an effective tool for such data augmentation. Author has primarily used Finetuned RoBERTa classifier and the task of harmful language detection to showcase the efficacy. Reviewers agree that the discussion and comparison around manual vs automated data augmentation is an interesting contribution. Most of the points such as manual evaluation of the generated data, concentrating only on RoBERTa has been addressed by the authors during rebuttal period. It is interesting work and the authors should incorporate the suggestions made by the reviewers to the best of their capability.